# A Critical Look at the Consistency of Causal Estimation with Deep Latent Variable Models

**Severi Rissanen**
Department of Computer Science
Aalto University
Espoo, Finland
`severi.rissanen@aalto.fi`

**Pekka Marttinen**
Department of Computer Science
Aalto University
Espoo, Finland
`pekka.marttinen@aalto.fi`

## Abstract

Using deep latent variable models in causal inference has attracted considerable interest recently, but an essential open question is their ability to yield consistent causal estimates. While they have demonstrated promising results and theory exists on some simple model formulations, we also know that causal effects are not even identifiable in general with latent variables. We investigate this gap between theory and empirical results with analytical considerations and extensive experiments under multiple synthetic and real-world data sets, using the causal effect variational autoencoder (CEVAE) as a case study. While CEVAE seems to work reliably under some simple scenarios, it does not estimate the causal effect correctly with a misspecified latent variable or a complex data distribution, as opposed to its original motivation. Hence, our results show that more attention should be paid to ensuring the correctness of causal estimates with deep latent variable models.

## 1 Introduction

Causal inference, dealing with the questions of when and how we can make causal statements based on observational data, has been a topic of growing interest in the deep learning community recently. On the one hand, causal inference promises to provide traditional machine learning and AI with methods for explainability, domain adaptation, and causal reasoning capabilities in general [Pearl, 2019]. On the other hand, many deep learning methods for improving causal inference have been proposed. Some of the models have been designed under the assumption of no unobserved confounding [Shalit et al., 2017, Yoon et al., 2018, Shi et al., 2019], while others utilize latent variables in one way or the other to account for unobserved confounders [Louizos et al., 2017, Rakesh et al., 2018, Pfohl et al., 2019, Madras et al., 2019, Mayer et al., 2020, Chen et al., 2020, Pawlowski et al., 2020, Jesson et al., 2020]. Although some simple models with unobserved confounders are known to produce correct results [Angrist et al., 1996, Pearl et al., 2016, Miao et al., 2018], the *consistency*, i.e., whether in the limit of large data the correct causal effect is retrieved, is usually left largely open with deep latent variable models.

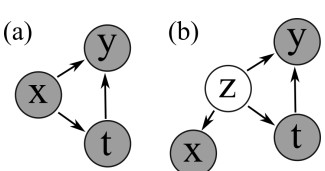

Figure 1: (a) The direct confounding causal graph. (b) The causal graph with an unobserved confounder $z$, and proxy variables $x$.

In particular, Louizos et al. [2017] proposed the causal effect variational autoencoder (CEVAE) for performing causal inference in the setting where we have an unobserved confounder, of which only noisy proxy variables are available. The causal graph is shown in Fig.1b, which contrasts to the standard "no unobserved confounding", or direct confounding, graph in Fig.1a. The proxy variable setting has been studied rigorously elsewhere by [Pearl et al., 2016], where the authors provided

35th Conference on Neural Information Processing Systems (NeurIPS 2021).

provably correct methods for a few simple types of data with strong assumptions about the data generating process. They dubbed the process of estimating causal effects in this context as "effect restoration". CEVAE was proposed to relax these assumptions significantly, but the question of consistency was left unanswered.

To provide insight into the behavior of deep latent variable models as causal effect estimators, we use CEVAE as a case study. CEVAE is a natural choice because it is based on the well-established standard variational autoencoder [Kingma and Welling, 2014, Rezende et al., 2014], and it allows comparison with analytical, provably correct methods. We conduct rigorous experiments with the model using various synthetic and semi-synthetic data sets and also provide theoretical statements in some special cases. Even though generic theoretical results are difficult to get for variational autoencoders, we provide intuitive conclusions on the assumptions under which the model works or does not work.

## 2 Preliminaries

### 2.1 Model description

The objective in the scenario in Fig.1b is to learn the causal effect between the variables $t$ and $y$. $t$ is a variable on which we can perform an intervention, e.g., a treatment on a disease. $x$ is the possibly multidimensional proxy providing indirect and noisy information about the unobserved confounder $z$. The correct interventional distribution $p(y|do(t))$ is defined with the formula

$$p(y|do(t)) = \int p(y|z,t)p(z)dz, \tag{1}$$

where the integral is replaced with a sum for a discrete confounder. Note that we denote the latent variable of a VAE with $z$ as well, even though they are conceptually separate and might not be distributed in the same way. The distinction should be clear from the context.

At its core, the causal effect variational autoencoder is simply a regular variational autoencoder with two additional assumptions. First, we assume that the latent variable $z$ corresponds to the unobserved confounder in some way so that after training, we can get the causal effect $p(y|do(t))$ by estimating with the adjustment formula:

$$p(y|do(t)) \approx p_\theta(y|do(t)) = \int p_\theta(y|z,t)p(z)dz, \tag{2}$$

where $\theta$ are learned decoder parameters and $p(z)$ is the VAE prior. Second, we assert some additional restrictions on the structure of the decoder. The idea is that the quantities reconstructed during training should follow the conditional independencies specified by the causal graph in Fig.1b so that the conditional probability of observed variables given the latent variable factorizes as

$$p_\theta(x^i, t^i, y^i|z) = p_\theta(x^i|z)p_\theta(t^i|z)p_\theta(y^i|z,t^i), \tag{3}$$

where superscript $i$ refers to the $i$:th observation. Thus, we can write the ELBO for the model as

$$\mathcal{L}(\theta, \phi) = \sum_i \big[ \mathbb{E}_{q_\phi(z|x^i,t^i,y^i)}[\log p_\theta(x^i|z) + \log p_\theta(t^i|z)$$
$$+ \log p_\theta(y^i|z,t^i)] - \mathrm{KL}[q_\phi(z|x^i,t^i,y^i)||p(z)]\big], \tag{4}$$

where $x^i, t^i$ and $y^i$ are observed quantities. Thus, we have to define at least four neural networks: The encoder network $q_\phi(z|x^i,t^i,y^i)$ and three decoder networks corresponding to the conditionals in Eq.3. The original paper also suggested composing the encoder of multiple networks chosen based on the value of the treatment, but that is not strictly necessary and was not motivated by the causal graph. Note that the decoder differs from usual VAE decoders in that the observed treatment value $t^i$ has to be given as input to the network corresponding to $p_\theta(y|z,t)$. The original paper seemed to suggest that the input should be sampled from the $p_\theta(t|z)$ distribution during training (Fig.2b in [Louizos et al., 2017]), but the factorization in Eq.3 suggests that the observed $t$ should be used instead.

In terms of the problem statement, CEVAE is perhaps most closely connected to the methods of effect restoration first proposed by Kuroki and Pearl [2014]. They offered provably correct, analytical solutions to the proxy variable problem in Fig.1b when all variables are jointly normal or categorical

and $x$ consists of two variables conditionally independent of each other given $z$. The setting was further studied in [Miao et al., 2018]. Note that since CEVAE assumes a different causal graph from the direct confounding graph in Fig.1a, it can't be assumed to give correct results if the true data generating process is the direct confounding graph. For example, Kuroki and Pearl [2014] showed that an observed $p(x, t, y)$ can map to two completely different causal effects by assuming either the direct or the unobserved confounding graphs. As the original CEVAE paper experimented on data with no unobserved confounding in addition to data sets with the unobserved confounding, we suspect that this may have been a point of confusion for some.

## 2.2 Possible issues with estimation

Theorem 1 in [Louizos et al., 2017] states that if CEVAE is able to recover the $p(z, x, t, y)$ distribution, it is guaranteed to yield correct causal estimates. However, this leaves open the relevant question about when such an assumption can hold, as recovering $p(z, x, t, y)$ entirely is, strictly speaking, impossible due to the unidentifiability of VAEs (e.g. [Locatello et al., 2019]). In contrast, we don't consider the latent variable of CEVAE to strictly correspond to the true confounder. Instead we view the process of training CEVAE and applying the adjustment formula in Equation 2 simply as a statistical estimator for the causal effect, regardless of what the latent variable exactly represents. That is, we are interested in recovering the correct causal effect and not the true hidden confounder. Our goal is then to study when the resulting causal effect estimates are *consistent*. Consistency means in general that the estimates of the parameters of interest approach their correct values as the amount of data increases, see, e.g., [Schervish, 2012].

For consistency to be possible, the model also has to be *identifiable* with respect to causal effect estimates, so that minimizing the loss (e.g. maximizing the likelihood or the ELBO) does not map to multiple data generating parameters $\theta$ that correspond to different $p_\theta(y|do(t))$, see, e.g., [Murphy, 2012]. This is not to be confused with the common usage of the word "identifiability" in causal inference literature, where the question is instead whether it is possible in principle to estimate some causal effect from an observed distribution, e.g., using the techniques of the famous *do-calculus* [Pearl, 2009]. In the presence of unobserved variables, results on causal identifiability often rely on parametric knowledge about the underlying data generating process. We refer to the identifiability of CEVAE as a statistical estimator as the *model identifiability*, to distinguish it from the causal identifiability. Note that model identifiability as defined here guarantees only that a unique causal effect estimate is obtained, but not necessarily that it is consistent, for example if the model is misspecified.

The original CEVAE paper suggested multiple scenarios as the motivation for the model, including the case where we have very few parametric assumptions about the data generating process and the case where the distribution of the proxies is complex, as is often reasonable to assume with real-world data. Our aim here is to study how well the model works as we reduce the number of assumptions and move further into the territory envisioned in the paper. Based on the motivating scenarios, we can conceptually separate three goals for CEVAE:

1. It should produce correct estimates with minimal knowledge about the parametric forms of the data generating $p(x|z)$, $p(t|z)$ and $p(y|z, t)$ distributions.

2. It should work if we don't know the form of the unobserved confounder's distribution, which could be categorical or Gaussian, for instance.

3. It should work with an arbitrarily complex distribution of proxies.

With the first goal, the hope is that neural networks will estimate the conditionals correctly enough to estimate the causal effects. With the second goal, the hope is that the true confounder is represented well enough with the standard Gaussian prior of the VAE.

In practice, correct estimation could be prevented by many factors. For real-world data sets, it is possible that the causal effect is not identifiable at all from the data, even if we know the parametric form of the data generating process. If it is identifiable in principle, CEVAE might still fail due to inherent model nonidentifiability caused by the nonparametric assumptions and because we don't have a guarantee that finding a unique global optimum for the ELBO leads to a correct causal estimate. Local minima or other issues with optimization could cause further practical problems with the correctness of the results.

We focus on the estimation of $p(y|do(t))$ instead of individual-level causal effects $p(y|do(t), x)$ as that makes the analysis more straightforward. Note that while the average treatment effect (ATE), defined as $\mathbb{E}[y|do(t=1)] - \mathbb{E}[y|do(t=0)]$, is a common metric, we are interested in $p(y|do(t))$ directly because the estimated ATE can have the correct value even when the estimates of $p(y|do(t=1))$ and $p(y|do(t=0))$ are not correct, and furthermore our analysis extends to continuous $t$.

## 3 Results

### 3.1 Setup with provably identifiable simple synthetic data

As a first step, we studied the simplest possible cases where we know from theory that the causal effects are identifiable in principle and provably correct analytical estimation methods exist, i.e., the two data types discussed in [Kuroki and Pearl, 2014]. Aside from being important basic cases, these data sets are interesting because they are relatively simple to study, we can compare CEVAE results to the analytical methods, and we can try to extract some qualitative understanding from the results.

**Linear-Gaussian data** The first data type was the linear-Gaussian, where the true data generating distribution is such that all variables are jointly normally distributed, but respecting the conditional independences of graph 2 in Fig.1b. The proxy $x$ also consists of two variables $x_1$ and $x_2$ that are conditionally independent given z:

$$z \sim N(0,1), x_1|z \sim N(c_1 z, \sigma_{x_1}), x_2|z \sim N(c_2 z, \sigma_{x_2})$$
$$t|z \sim N(c_t t, \sigma_t), y|z, t \sim N(c_{yz} z + c_{yt} t, \sigma_y)$$

Here, $c_i$ and $\sigma_i$ are predefined parameters. To avoid jumping to conclusions based on arbitrary choices of data generating parameters, we sampled them randomly from a distribution that should provide a wide range of different generating processes. We detail the sampling method in the Supplementary Material.

**Binary data** In the other type of data, all variables were binary, and in particular, $x$ consisted of two binary variables $x_1$ and $x_2$ that were conditionally independent given $z$. We sampled the data generating process from a distribution explained in the Supplementary Material. The binary data also tests the ability of CEVAE to perform correctly even if the assumption of a normally distributed unobserved confounder is not valid.

To estimate the correctness of causal effect estimates using neural networks with the linear-Gaussian data, standard metrics such as ATE error do not apply because the treatment variable is not binary. Instead, we define the *Average Interventional Distance* (AID):

$$\text{AID} = \int p(t) \int |p_\theta(y|do(t)) - p(y|do(t))| dy dt \tag{5}$$

where the integrals can be changed to sums for discrete variables. In addition to being defined for continuous treatments, this metric has the advantage that it will only approach zero if $p_\theta(y|do(t))$ approaches $p(y|do(t))$ for all values of $t$ that we have data from, and it follows the intuition that we should be more confident for values of $t$ for which we have lots of data.

**Estimation models** In the experiments, we refer to the "full CEVAE" as a model where all conditional distributions are parameterized with three-layer MLPs with layer width 30 and a latent variable dimension of 10. With the linear-Gaussian data, the standard deviations of the Gaussian conditionals were individually estimated for each data point, using the standard assumption of diagonal covariance in the encoder and decoder. For the linear-Gaussian data, we also considered the "linear CEVAE", a model with conditional distributions represented by simple linear layers and single standard deviations, shared between all data points and estimated for each conditional. We tried latent dimensionalities of 1, 2, and 10 for these linear models. Further details are provided in the Supplementary Material.

### 3.1.1 Results with linear-Gaussian data

It is, in fact, possible to show that the one-dimensional linear CEVAE is consistent in this situation, as encapsulated in the following proposition:

**Proposition 1.** *A linear CEVAE with a one-dimensional latent space estimates the causal effect correctly, given that it reaches the global optimum of the ELBO with infinite data.*

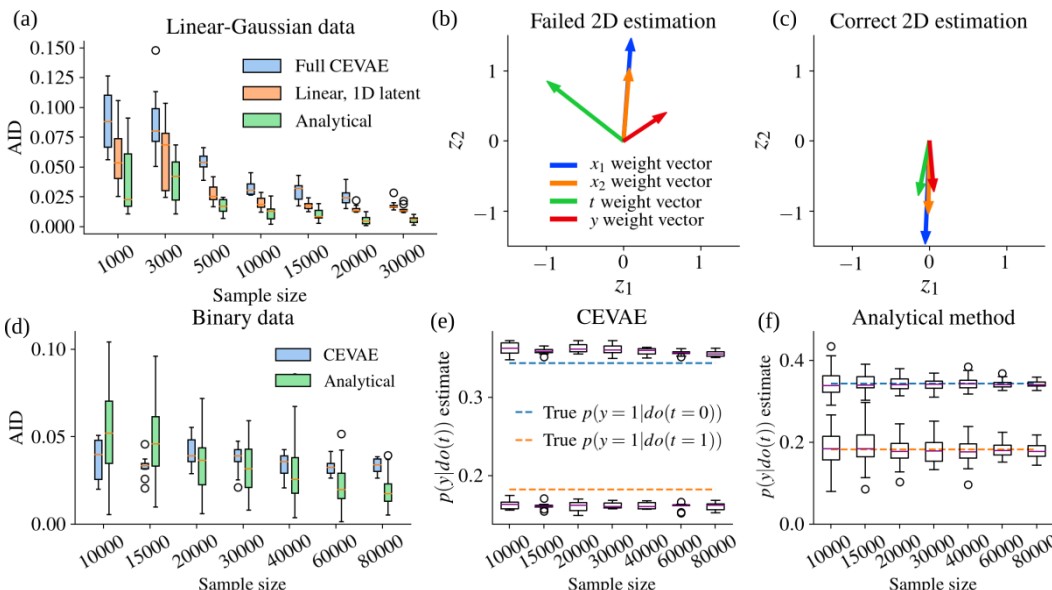

Figure 2: Top row, linear-Gaussian generative model: (a) AID values for the full CEVAE (NN conditionals and 10D latent space), for the simpler CEVAE with linear conditionals and 1D latent space, and for the analytical method. (b) Weight vectors of the different conditional distribution mean functions for a linear CEVAE with a two-dimensional latent space. Here parameters were initialized manually, and estimation failed. (c) The same parameter estimates for a model that estimated causal effects correctly. Here, only one latent dimension is used. Bottom row, binary generative model: (d) AID values for the full CEVAE and the analytical method with respect to sample size. (e) Estimates of the $p(y|do(t))$ values for full CEVAE. (f) The same estimates with the analytical method.

The proof, which relies on the earlier result by [Kuroki and Pearl, 2014], is provided in the Supplementary Material. Thus, a well-specified CEVAE model can result in correct estimation. The question then remains whether overparameterization by neural networks breaks the consistency.

Figure 2a shows the AID of the different models and the analytical method of [Kuroki and Pearl, 2014] as a function of sample size, for one data generating distribution. Analytical estimates for some of the required parameters for calculating $p(y|do(t))$ were not provided in the original paper, but we derive them in the Supplementary Material. While the analytical method and the linear CEVAE with a 1D latent variable perform better than the full CEVAE, all of them seem to converge towards the correct $p(y|do(t))$ distribution. We show in the Supplementary Material that the result is robust, as the estimates converge to the correct distribution for other data-generating parameters. Thus, overparameterizing the conditional distributions with NNs or using a larger than required latent variable dimension doesn't necessarily break the estimation of the causal effect.

To highlight that the result is not obvious, we ran additional experiments with a model using linear conditional distributions but with a two-dimensional latent space, i.e., with one redundant dimension. With some initializations, the model ended up estimating the causal effect incorrectly, but with an indistinguishable ELBO compared to a model with the correct causal effect. The initialization and other details are given in the Supplementary Material. Figure 2b visualizes the minimum with the wrong result. Essentially, the model uses only one latent dimension to reconstruct the proxies $x$ while treatment $t$ and outcome $y$ are reconstructed partly with the other dimension as well, preventing correct modeling of dependencies between observed variables. In practice, with a random initialization, this happens only rarely and not at all with the full CEVAE due to the tendency of *posterior collapse* in VAEs, causing the model to use one dimension only. An example for the 2D linear CEVAE is shown in Fig.2c. Hence, whereas the posterior collapse is often an unwanted characteristic of VAEs [He et al., 2018, Razavi et al., 2019, Dai et al., 2020], it here seems to save the day, although an unnecessarily high latent dimension could still cause issues in principle. In the Supplementary Material, we visualize the posterior collapse phenomenon for the full CEVAE and

also show that the 10D linear CEVAE causal effect estimates become systematically incorrect if we prevent posterior collapse using KL divergence annealing.

**Conclusion** Overparameterizing the conditionals with neural networks does not necessarily prevent correct estimation, but a too high-dimensional latent variable could in principle. Posterior collapse usually resolves the problem, however.

### 3.1.2 Results with binary data

The AID values and corresponding causal effect estimates for the full CEVAE and the analytical method by [Miao et al., 2018] are plotted as a function of sample size in Fig.2d. CEVAE, which incorrectly assumes a Gaussian latent variable, produces reasonable results but fails to converge to the correct causal effect. In contrast, when using the analytical method, the estimate gets better and better as the sample size increases. We show in the Supplementary Material that similar results are obtained for different data generating distributions. Note that the binary data generating process is possibly the simplest process where the actual confounder is not normally distributed and the causal effects are identifiable in principle. We conclude that CEVAE does not, in general, estimate the causal effect correctly when the latent variable is not specified appropriately in advance, and the second goal of CEVAE, mentioned in Sec.2.2, is not met. In the Supplementary Material, we also show that a version of CEVAE with a binary latent variable can produce correct causal estimates, although it is prone to get stuck in local minima for some data sets.

**Conclusion** CEVAE does not estimate causal effects correctly if the latent variable is misspecified, in general.

## 3.2 Illustration of difficulties with complex data

This section describes two additional experiments with synthetic data, illustrating issues that can reasonably be assumed to come up with real data as well. The causal effects are identifiable with these data sets because they are based on the linear-Gaussian data. The estimation model is the same as the full CEVAE as specified in the previous section unless mentioned otherwise.

### 3.2.1 Data with irrelevant variation in the proxies

The former experiments with linear-Gaussian data showed that an unnecessarily high-dimensional latent variable could cause issues, especially with the simpler linear CEVAE which did not exhibit posterior collapse, but the problem could be avoided by using a 1D latent. To see how a higher-dimensional latent variable could be necessary with complex data, we used the linear-Gaussian data set but added a third proxy variable that contained irrelevant, high-variance noise. After generating the proxies, an additional rotation was applied to them in the three-dimensional space so that the relevant variation was "hidden" in a specific two-dimensional subspace. The process is illustrated in Fig.3a. Here, with the standard assumption of diagonal covariance in the decoder, we can expect a 1D CEVAE to focus on modeling the noise because that dominates the loss. In contrast, a higher-dimensional CEVAE has the option to explain the noise with one and the signal with another dimension, enabling correct inference.

Figure 3b shows the full CEVAE AID as a function of sample size for one- and two-dimensional latent variables. As expected, the model with a two-dimensional latent variable estimates the causal effect correctly, while the one-dimensional model does not.

**Conclusion** A high-dimensional latent variable can be necessary to estimate the causal effect correctly using very noisy proxies, even if the actual confounder is one-dimensional.

### 3.2.2 Data with repeated proxies

The second difficulty in complex data can arise when the proxies contain variation relevant to predicting $t$ and $y$, but there exist significant correlations in $x$ that are not caused by the confounder. This can result in the model adjusting directly to $x$, ignoring the unobserved confounder entirely. As an example, consider the linear-Gaussian data set, but with a modification that we add two proxies, $\tilde{x}_1$ and $\tilde{x}_2$, which are copies of the original two. Here, the following proposition holds:

**Proposition 2.** *With an altered linear-Gaussian data generating process where we have additional proxies $\tilde{x}_1 = x_1$ and $\tilde{x}_2 = x_2$, the value of the ELBO of a 2D CEVAE can approach infinity while the*

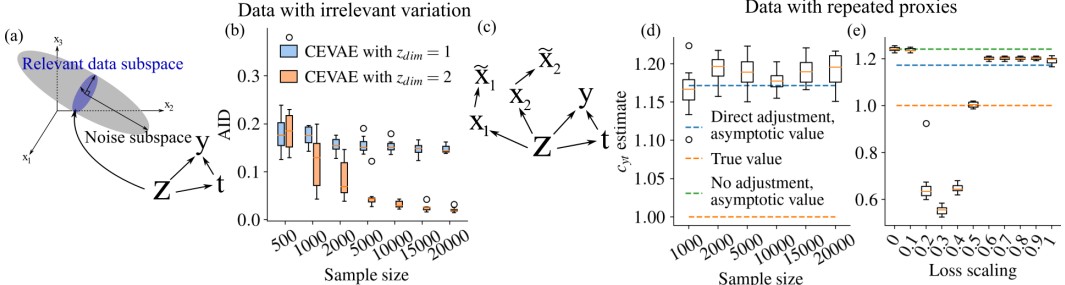

Figure 3: (a) Linear-Gaussian data generating process with irrelevant noise and a "hidden" relevant subspace. (b) AID values with respect to sample size for models using 1D and 2D latent spaces for the data with irrelevant noise. (c) Data generating process for linear-Gaussian data with copies $\tilde{x}_1$ and $\tilde{x}_2$. (d) Linear, 1D CEVAE $c_{yt}$ estimate as a function of sample size for the linear-Gaussian data with copies. (e) The $c_{yt}$ estimates as a function of proxy loss scaling for the linear 1D CEVAE.

*causal effect estimate converges to the value that is obtained by adjusting directly to the proxies, that is, $\int p(y|x,t)p(x)dx$.*

The proof is included in the Supplementary Material. The intuition is that whenever the model reconstructs one of the proxies, it can easily reconstruct the copy as well with the same accuracy, effectively doubling the importance of the proxy reconstruction loss. If we use the latent representation to directly represent the proxies with increasing accuracy, the negative KL divergence term in the ELBO decreases slower than the proxy reconstruction term increases, and the ELBO approaches infinity. At the same time, the latent space becomes a representation of the proxies, and the $y$ reconstruction term in the ELBO forces the corresponding predictor $p_\theta(y|z,t)$ to become an approximation of $p(y|x,t)$. We hypothesize that the same phenomenon will be an issue with a complicated distribution of proxies since, most likely, there will be similar correlations that are not directly caused by the unobserved confounder and that can cause the model to focus too much on proxy reconstruction.

As a simple empirical demonstration of this phenomenon in a more realistic distribution, we experimented with data where a small Gaussian noise is added to $\tilde{x}_1$ and $\tilde{x}_2$ so that the correlations with $x_1$ and $x_2$ are not perfect. We modified the full CEVAE to use a linear predictor for the conditional distribution of $y$ to make the results easier to interpret (the estimated regression coefficient of $t$ should approach the $c_{yt}$ coefficient in the data generating process, i.e., the true causal effect). Figure 3d shows the $c_{yt}$ estimate as a function of sample size. As expected, the estimate corresponds to the direct adjustment value that we would get if we used linear regression to predict $y$ based on $x$ and $t$.

Given that the loss function with repeated proxies corresponds to the regular loss where the proxy reconstruction term is multiplied by a factor of two, the most obvious way to resolve the problem is to adjust the reconstruction loss manually by a factor of one-half. In general, we would scale the term $\mathbb{E}_{q_\phi(z|x^i,t^i,y^i)}[\log p_\theta(x^i|z)]$ in Eq.4 by some factor $\lambda < 1$, forcing the VAE to put less weight on just reconstructing the proxies. However, in a real-world situation, it is not obvious what this scaling factor should be. To illustrate the effect of the scaling factor, Figure 3e shows the $c_{yt}$ estimate as a function of proxy loss scaling for a sample size of 20000. With scaling factors close to one, the results are close to the direct adjustment results for the reasons explained. When lowering it to one-half, the estimate abruptly changes and ends up in the true value, as expected. When lowering it further, however, the results change as well and we start getting incorrect values. As the scaling factor approaches zero, CEVAE stops using the proxy data at all, and our estimate for $c_{yt}$ becomes equal to the one we would get by assuming $p(y|do(t)) = p(y|t)$, i.e., no confounding. In the Supplementary Material, we prove the following proposition, which states that this estimate corresponds to many global optima of the ELBO where the latent space is either neglected entirely or not used in the reconstruction of either $t$ or $y$:

**Proposition 3.** *With the proxy reconstruction loss scaled to zero, one set of global optima to the CEVAE ELBO is such that $p_\theta(y|do(t)) = p(y|t)$ and either $t$ or $y$ is not dependent at all on the latent variable for the linear-Gaussian data.*

The intuition is that if one of the causal links $z \to t$ or $z \to y$ is removed in CEVAE, then the $c_{yt}$ estimate is produced as if there was no confounding. While other global optima exist, it seems that we get these no-adjustment solutions in practice.

**Conclusion** CEVAE can overemphasize modeling of the proxies with some data sets, leading it to ignore the unobserved confounder entirely. We may be able to fix this by scaling the reconstruction loss for proxies, but it is not clear how to choose the scaling in practice.

### 3.3 Semi-synthetic data

Here we describe two experiments based on real-world data sets. Details on the experimental setup, data generating processes, and neural network architectures are provided in the Supplementary Material. Here, we don't have guarantees that the causal effects are identifiable in principle from the data, which corresponds to the real-world situation where we can never be sure about identifiability without access to the parametric form of the data generating process. In any case, the experiments allow us to highlight issues with estimation failure that are not related to the identifiability of the data.

#### 3.3.1 Proxy IHDP data set

We decided to investigate the performance of CEVAE on a modified version of the Infant Health and Development Program (IHDP) data set [Hill, 2011], which was created from a study on the effects of intensive child care on future test scores of premature infants [Brooks-Gunn et al., 1992]. It consists of 25 covariates, some continuous and some categorical, treatments, and synthetically generated test scores. The IHDP data is a well-known causal inference benchmark, but as such, it is not suitable for our study because the $y$ values in the data have been generated directly based on the covariates, and thus the data doesn't follow the causal graph assumed by CEVAE. To overcome this, we singled out one of the covariates to be the hidden confounder, leaving the rest as proxies. The treatment $t$ and the recovery $y$ were then generated using the chosen confounder. Technically speaking, we don't know the direction of causalities between the confounder and the proxies defined this way, but the data is still Markov equivalent to the graph in Fig.1b. Since the original data was very small (just 747 subjects), we trained a variational autoencoder on the covariates to generate more data from a distribution that is similar to the original one and which should be realistic enough for our purposes. The generated data set has the benefit that the distribution of the unobserved confounder and the proxies is not arbitrarily defined, instead following a real-world distribution that is relevant for causal inference. In the Supplementary Material, we show that the unobserved confounder is clearly correlated with many of the proxies, and thus it's possible that the proxies provide us enough information to make the causal effects identifiable in principle.

Figure 4a shows that the causal estimate does not approach the correct value as we increase the sample size, instead corresponding to the value we would get from direct adjustment to proxies, similarly to Section 3.2.2. Note that although we don't have strict guarantees that the causal effects are identifiable with this data set, the result nevertheless shows that the problem of placing too much weight on proxy reconstruction is relevant in a realistic use-case as well. In Fig.4b we investigate whether scaling the proxy loss can recover the correct causal effect. The pattern is similar as before: Scaling with a factor close to one recovers direct adjustment. With a scaling factor close to zero, $y$ is predicted solely based on $t$. With intermediate scaling, some estimates are approximately correct, but this time not consistently for any of the scaling factors.

**Conclusion** The problem of adjusting directly to the proxies, described in Sec.3.2.2, happens with real data.

#### 3.3.2 Proxy MNIST data set

In this section, we experiment with a data set where the proxies are images, which is a natural domain for neural network based models such as CEVAE. To create data that follows the correct causal graph, we trained a GAN with a three-dimensional latent variable on the MNIST data [LeCun and Cortes], after which we used the GAN to generate a value between zero and one for each pixel given the latent value. We interpreted these as probabilities and sampled each pixel from the corresponding Bernoulli distribution to generate noisy images, which were used as the proxies. We used the first latent dimension of the GAN as the unobserved confounder $z$ and generated binary $t$ and $y$ values based on the chosen $z$ for each sampled image. We also generated an additional linear-Gaussian

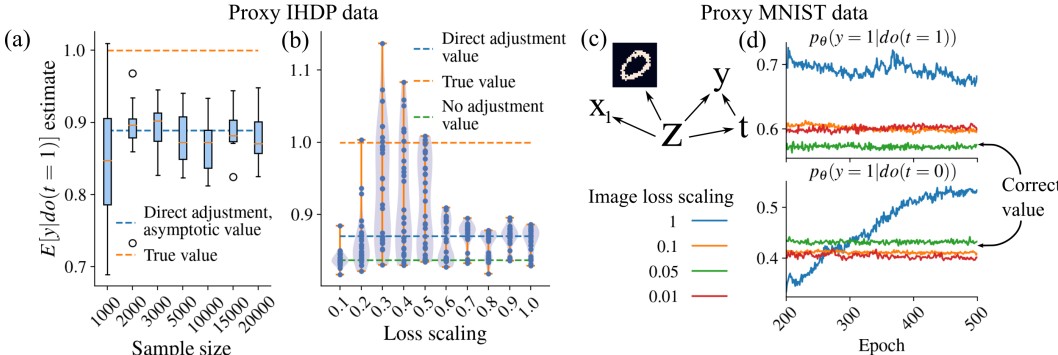

Figure 4: (a) $\mathbb{E}[y|do(t=1)]$ estimates for the IHDP data with respect to sample size. (b) $\mathbb{E}[y|do(t=1)]$ estimates for the IHDP data as a function of proxy loss scaling, for a data set of size 20000. The figures with $\mathbb{E}[y|do(t=0)]$ are included in the Supplementary Material. (c) The data generating process for the proxy MNIST data set. (d) Causal effect estimates for different image loss scaling values with respect to training time, for a data set of size 10000.

proxy $x_1$ to increase the chance that the causal effects are identifiable in principle, while not reducing the problem to trivial. The created data set, illustrated in Fig.4c, then followed the correct causal graph with a normally distributed unobserved confounder.

Since the image data is very complex and high-dimensional, it is reasonable to expect that CEVAE can put too much weight on image reconstruction in one way or another during training. Thus, we experimented by scaling the reconstruction loss term of the images to different values. Indeed, figure 4d shows us that without any scaling, the estimated $p(y|do(t))$ values do not converge to anything even with 500 epochs. This is possibly due to the reconstruction term being much larger than the rest, so that fluctuations there overshadow the modeling of the other variables. When scaling the image loss to lower values, the estimates start to converge but are not quite correct for values 0.1 and 0.01. For scaling value 0.05, however, we recover almost exactly the correct result. The result is confirmed in the Supplementary Material with AID values from multiple runs. Thus, it seems that scaling the loss function appropriately can result in the correct causal effect even with data as complex as images, if we know the correct scaling.

**Conclusion** With very complex proxy data, getting an estimate for the causal effect can be difficult. In some cases, scaling the loss appropriately can result in correct estimation even with real data.

### 3.3.3 Twins data set

We also ran experiments on the Twins data set presented in the original paper [Louizos et al., 2017, NCHS, 1996]. It provides an example where the causal effects are identifiable in principle, the confounder is categorical, and where the distribution of $z,t$, and $y$ is based on real-world data. In the Supplementary Material, we show that CEVAE doesn't return consistent estimates, further reinforcing our conclusion about incorrect estimates with misspecified latent variables.

## 4 Discussion

Two of the goals we listed for CEVAE in Section 2.2 were that it should recover causal effects even if we don't know the distribution of the confounder and if the distribution of the proxies is complex. It appears that CEVAE does not work consistently correctly in either case. Thus, while using a deep latent variable model in this context shows some promise, new solutions are needed to overcome the issues that come up with real data. Although there is an absence of theory supporting the model in general, these results were non-obvious to us and we believe that they are useful for many others, given the large amount of research in the field. The main limitation of our work is that we focused on CEVAE, but we believe that the qualitative results and recognized problems will be useful in research on other, similar models. Another limitation is that our study was mainly empirical and we can not provide theoretical guarantees that the qualitative results transfer to all possible data sets. The

negative results do, however, serve as counterexamples, and we did attempt to provide intuition for the phenomena observed, allowing future researchers to assess whether our work is relevant for them.

Finally, our aim is not to discourage research with CEVAE or deep latent variable models for causal inference in general, but instead, we hope that our results will accelerate progress in the field. The hope is that our results are a starting point for thinking about the consistency of causal estimation with similar models and advancing guarantees for it. In any case, our experiments showed that we should not brush off the issue entirely since the alternative is that the model can produce incorrect results even with some rudimentary data sets.

## 5  Impact Statement

If CEVAE and similar machine learning based methods for causal inference become usable and prevalent in application areas such as epidemiology and the social sciences, this line of work could have a clear positive social impact by enabling research in these fields with new possibilities and thus allowing for better decision-making through better understanding of important phenomena. On the other hand, it's possible that practitioners become overly reliant on the claims that these methods estimate causal effects with very few assumptions, becoming less rigorous in considering the assumptions that are necessary to make (e.g., the causal graph). This could inadvertently have the negative impact of degrading the quality of research. Researchers in machine learning and causal inference should strive to avoid this by communicating the limitations realistically to practitioners and make the necessary assumptions as explicit as possible.

## Acknowledgements

PM has received funding from the Academy of Finland (grants 336033, 315896), BusinessFinland (grant 884/31/2018), and EU H2020 (grant 101016775).

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
