# A Critical Look at the Consistency of Causal Estimation with Deep Latent Variable Models: Supplementary Material

## A  Analytical estimate of $p(y|do(t))$ for linear-Gaussian data

In this section, we derive an analytical estimate of the parameters of the $p(y|do(t))$ distribution for the linear-Gaussian data, including the ones that weren't provided in the original paper. It assumes that we know the parametric form of the generating process. The approach is slightly different from the original paper, which utilized higher-level properties of the structural model in their derivation. We will first derive the asymptotic, infinite data covariance matrix of the observed variables expressed using the data generating parameters $c_1, c_2, \sigma_1$, etc., after which we can derive expressions for the parameters using observable covariances. The formulas can then be used as asymptotically correct estimates with finite data as well.

We start by finding out a form for the joint distribution, including $z$:

$$p(z, x, t, y) = p(z)p(x|z)p(t|z)p(y|z, t)$$
$$\sim e^{-\frac{1}{2}\left(z^2 + \frac{(c_1 z - x_1)^2}{\sigma_1^2} + \frac{(c_2 z - x_2)^2}{\sigma_2^2} + \frac{(c_t z - x_t)^2}{\sigma_t^2} + \frac{(c_{yz} z + c_{yt} t - y)^2}{\sigma_y^2}\right)}$$

This is a jointly Gaussian distribution. We can find the covariance matrix by looking at the exponent and rearranging terms:

$$-2\log p(z, x, t, y) \sim z^2\left(1 + \frac{c_1^2}{\sigma_1^2} + \frac{c_2^2}{\sigma_2^2} + \frac{c_t^2}{\sigma_t^2} + \frac{c_{yz}^2}{\sigma_y^2}\right) + z x_1\left(-\frac{2c_1}{\sigma_1^2}\right) +$$

$$z x_2\left(-\frac{2c_2}{\sigma_2^2}\right) + zt\left(-\frac{2c_t}{\sigma_t^2} + \frac{2c_{yz}c_{yt}}{\sigma_y^2}\right) + zy\left(-\frac{2c_{yz}}{\sigma_y^2}\right) + x_1^2\frac{1}{\sigma_1^2} +$$

$$x_2^2\frac{1}{\sigma_2^2} + t^2\left(\frac{1}{\sigma_t^2} + \frac{c_{yt}^2}{\sigma_y^2}\right) + ty\left(-\frac{2c_{yt}}{\sigma_y^2}\right) + y^2\left(\frac{1}{\sigma_y^2}\right)$$

$$= \begin{bmatrix} z & x_1 & x_2 & t & y \end{bmatrix} C^{-1} \begin{bmatrix} z \\ x_1 \\ x_2 \\ t \\ y \end{bmatrix}$$

where

$$C^{-1} = \begin{bmatrix} 1 + \frac{c_1^2}{\sigma_1^2} + \frac{c_2^2}{\sigma_2^2} + \frac{c_t^2}{\sigma_t^2} + \frac{c_{yz}^2}{\sigma_y^2} & -\frac{c_1}{\sigma_1^2} & -\frac{c_2}{\sigma_2^2} & -\frac{c_t}{\sigma_t^2} + \frac{c_{yz}c_{yt}}{\sigma_y^2} & -\frac{c_{yz}}{\sigma_y^2} \\ -\frac{c_1}{\sigma_1^2} & \frac{1}{\sigma_1^2} & 0 & 0 & 0 \\ -\frac{c_2}{\sigma_2^2} & 0 & \frac{1}{\sigma_2^2} & 0 & 0 \\ -\frac{c_t}{\sigma_t^2} + \frac{c_{yz}c_{yt}}{\sigma_y^2} & 0 & 0 & \frac{1}{\sigma_t^2} & -\frac{c_{yt}}{\sigma_y^2} \\ -\frac{c_{yz}}{\sigma_y^2} & 0 & 0 & -\frac{c_{yt}}{\sigma_y^2} & \frac{1}{\sigma_y^2} \end{bmatrix}$$

Inverting this, we get $C$, and the covariance matrix $C_{xty}$ of the marginal distribution $p(x, t, y)$ is got by dropping the row and columns corresponding to $z$, since $p(z, x, t, y)$ is jointly Gaussian:

$$C_{xty} = \begin{bmatrix} c_1{}^2 + \sigma_1{}^2 & c_1 c_2 & c_1 c_t & c_1(c_t c_{yt} + c_{yz}) \\ c_1 c_2 & c_2{}^2 + \sigma_2{}^2 & c_2 c_t & c_2(c_t c_{yt} + c_{yz}) \\ c_1 c_t & c_2 c_t & c_t{}^2 + \sigma_t{}^2 & c_t{}^2 c_{yt} + c_t c_{yz} + c_{yt}\sigma_t{}^2 \\ c_1(c_t c_{yt} + c_{yz}) & c_2(c_t c_{yt} + c_{yz}) & c_t{}^2 c_{yt} + c_t c_{yz} + c_{yt}\sigma_t{}^2 & c_t{}^2 c_{yt}{}^2 + 2c_t c_{yt}c_{yz} + c_{yt}{}^2\sigma_t{}^2 + c_{yz}{}^2 + \sigma_y{}^2 \end{bmatrix}$$

We then have a system of 10 equations, where each of the matrix cells corresponds to an asymptotic, infinite-data covariance. The equations of interest to us are

$$c_1 c_2 = \text{Cov}(x_1, x_2), \quad c_1 c_t = \text{Cov}(x_1, t), \quad c_2 c_t = \text{Cov}(x_2, t), \quad c_t^2 + \sigma_t^2 = \text{Var}(t)$$

$$c_2(c_t c_{yt} + c_{yz}) = \text{Cov}(x_2, y), \quad c_t{}^2 c_{yt} + c_t c_{yz} + c_{yt}\sigma_t{}^2 = \text{Cov}(t, y)$$

$$c_t{}^2 c_{yt}{}^2 + 2c_t c_{yt}c_{yz} + c_{yt}{}^2\sigma_t{}^2 + c_{yz}{}^2 + \sigma_y{}^2 = \text{Var}(y)$$

These can be solved to get

$$c_{yt} = \frac{\text{Cov}(t, y)\text{Cov}(x_1, x_2) - \text{Cov}(x_2, y)\text{Cov}(x_1, t)}{\text{Var}(t)\text{Cov}(x_1, x_2) - \text{Cov}(x_1, t)\text{Cov}(x_2, t)} \tag{1}$$

$$c_{yz}^2 = \frac{\text{Cov}(x_1, t)\text{Cov}(x_1, x_2)(\text{Cov}(t, y)\text{Cov}(x_2, t) - \text{Var}(t)\text{Cov}(x_2, y))^2}{\text{Cov}(x_2, t)(\text{Var}(t)\text{Cov}(x_1, x_2) - \text{Cov}(x_1, t)\text{Cov}(x_2, t))^2} \tag{2}$$

$$c_t^2 = \frac{Var(t)\text{Cov}(x_2, t)}{\text{Cov}(x_1, x_2)} \tag{3}$$

$$\sigma_t^2 = \text{Var}(t) - c_t^2 \tag{4}$$

$$c_t c_{yz} = \text{Cov}(t, y) - c_{yt}\sigma_t^2 - c_t^2 c_{yt} \tag{5}$$

$$\sigma_y^2 = \text{Var}(y) - c_{yz}^2 - c_{yt}^2\sigma_t^2 - 2c_{yt}c_t c_{yz} - c_t^2 c_{yt}^2 \tag{6}$$

where earlier expressions can be plugged in to later ones (especially $\sigma_y^2$ doesn't simplify much). The quantities $c_{yt}, c_{yz}^2$ and $\sigma_y^2$ are enough to characterize $p(y|do(t))$, since

$$p(y|do(t)) = \int_{-\infty}^{\infty} \frac{1}{\sqrt{2\pi}} e^{-\frac{z^2}{2}} \frac{1}{\sqrt{2\pi}\sigma_y} e^{-\frac{(y - c_{yz}z - c_{yt}t)^2}{2\sigma_y}} \, dz$$

$$= \frac{1}{\sqrt{2\pi}\sigma_{y|do(t)}} e^{-\frac{(y - \mu_{y|do(t)})^2}{2\sigma_{y|do(t)}}} \tag{7}$$

where

$$\mu_{y|do(t)} = c_{yt}t \tag{8}$$

$$\sigma_{y|do(t)} = \sqrt{\sigma_y^2 + c_{yz}^2} \tag{9}$$

In practice, we can use the asymptotically correct equations as formulas for estimation with finite data. The difference is that we use sample covariances and variances, and the parameter estimates are naturally correct only with infinite data.

## B  Proof of Proposition 1: The 1D linear CEVAE is consistent with linear-Gaussian data

In section A we showed that the causal effect $p(y|do(t))$ is identifiable from linear-Gaussian data, and presented an asymptotically correct, analytical method for estimation. Here we consider the 1D linear CEVAE, which estimates the conditional distributions linearly and has a latent dimension of one, thus being parameterized in the same way as the data generating process. We show that it is guaranteed to estimate the correct causal effect as well, assuming that we find the global optimum of the ELBO with infinite data. The proof relies on three facts:

1.  As shown in Sec.A, the parameters of the data generating process required for identifying the causal effect match one-to-one with observed covariances.

2. The CEVAE estimation model is defined so that the parameters match exactly with the data generating parameters and the prior is correctly specified as well.

3. The variational approximation to the posterior distribution can correctly represent the true posterior in this case.

**Proof.** We denote $p_\theta(\cdot)$ as the distribution induced by the VAE, e.g. $p_\theta(z, x, t, y) = p_\theta(x_1|z)p_\theta(x_2|z)p_\theta(t|z)p_\theta(y|z,t)p(z)$, where $p(z)$ is the zero mean, unit variance prior of the VAE.

Since the joint distribution $p_\theta(z, x, t, y)$ induced by the model is jointly Gaussian with mean zero, we know from properties of multivariate normal distributions that $p_\theta(z|x, t, y)$ is Gaussian as well with a mean that is a linear function of $x_1$, $x_2$, $t$ and $y$ with zero bias and constant variance. Thus, as we set our variational approximation $q_\phi(z|x, t, y)$ to be similarly a Gaussian with mean being a linear function of the observed variables and estimate the variance as one shared parameter, it can represent the true posterior $p_\theta(z|x, t, y)$ with the right choice of parameters $\phi$. Thus, the global optimum of the ELBO also equals the global optimum of the marginal log-likelihood. In the limit of infinite data, maximizing the sum of marginal log-likelihoods becomes equivalent to maximizing

$$\int p(x, t, y) \log p_\theta(x, t, y)\mathrm{d}z = \int p(x, t, y) \log \left( \frac{p_\theta(x, t, y)}{p(x, t, y)} p(x, t, y) \right) \mathrm{d}x\mathrm{d}t\mathrm{d}y$$

$$= -KL[p(x, t, y)||p_\theta(x, t, y)] + \int p(x, t, y) \log p(x, t, y)\mathrm{d}x\mathrm{d}t\mathrm{d}y \qquad (10)$$

where $p(x, t, y)$ is the true distribution of the data. Thus, since the parameter space of the linear VAE includes the true distribution, at the globally optimal $(\theta, \phi)$ combination the KL divergence goes to zero and $p_\theta(x, t, y) = p(x, t, y)$, i.e., the marginal distribution of our model is the true distribution. Because the VAE parameterization was defined in the exact same way as the generative model, we can then go through the exact same steps as we did in Sec.A, and notice that the estimate of $p(y|do(t))$ has to be the one we get from the true distribution. Thus, the model estimates $p(y|do(t))$ correctly. $\square$

## C  Proof of Proposition 2: We can get an infinite ELBO with copied proxies

In this section, we prove that we can get an infinite ELBO by using the latent space solely to reconstruct the proxies for linear-Gaussian data where the proxies are copied at least once. The proof assumes a linear CEVAE estimation model with a latent dimension of at least two, but the result is valid for a neural network parameterized CEVAE as well assuming that it can represent the same conditional distributions as the linear CEVAE. Given the universal approximation capabilities of neural networks, this is not a very radical assumption to make. The central idea in the proof is that we find a certain path in the parameter space which we then show to lead to an infinite evidence lower bound. It is constructed by mapping each value of $x$ to a corresponding position in the latent space, after which we let the encoder and decoder variances go to zero, forcing the reconstruction to become perfect.

**Proof.** Recall that the ELBO for CEVAE can be written in the form

$$\mathcal{L}(\theta, \phi) = \sum_i \left[ \mathbb{E}_{q_\phi(z|x^i,t^i,y^i)}[\log p_\theta(x^i|z) + \log p_\theta(t^i|z) + \log p_\theta(y^i|z,t^i)] - \right.$$

$$KL[q_\phi(z|x^i, t^i, y^i)||p(z)]] \qquad (11)$$

Let us now consider the scenario where we are trying to estimate the causal effect from a data set containing $N$ copies of the original two proxies. We restrict the analysis to the part of the parameter space where the variational approximation depends only on the proxies and both proxies are reconstructed using only one of the dimensions:

$$q_\phi(z|x, t, y) = \mathcal{N}(z|\mu_{z|x}, S_{z|x}) \qquad (12)$$

$$\mu_{z|x} = \begin{bmatrix} \gamma_{z_1} x_1 \\ \gamma_{z_2} x_2 \end{bmatrix} \qquad (13)$$

$$S_{z|x} = \begin{bmatrix} s_{z_1|x}^2 & 0 \\ 0 & s_{z_2|x}^2 \end{bmatrix} \qquad (14)$$

The proxy distribution in the decoder is set to $p_\theta(\{x_i\}|z) = \mathcal{N}(x_i|\gamma_i z, s_i^2)^N$, where $\gamma_i$ and $s_i$ are shared parameters for $x_i$ and all its copies, denoted by the set $\{x_i\}$ and $N$ is the number of copies $|\{x_i\}|$. Note that we use $\gamma$ and $s$ to highlight that these are parameters of CEVAE, not the data generating distribution, where we used $c$ and $\sigma$. Let's focus on the proxy reconstruction term and the KL divergence terms for the first latent dimension - proxy copy group pair, $z_1$ and $\{x_1\}$. The reconstruction term for a single observation is

$$\mathbb{E}_{q_\phi(z_1|x,t,y)}[\log p_\theta(\{x_1\}|z)]$$

$$= \int \frac{1}{\sqrt{2\pi}s_{z_1|x}} \exp(-\frac{(z_1 - \gamma_{z_1}x_1)^2}{2s_{z_1|x}^2}) \log\left[ \left( \frac{1}{\sqrt{2\pi}s_1} \exp(\frac{(x_1 - \gamma_1 z)^2}{2s_1^2}) \right)^N \right] dz \qquad (15)$$

$$= \int \frac{1}{\sqrt{2\pi}s_{z_1|x}} \exp(-\frac{(z_1 - \gamma_{z_1}x_1)^2}{2s_{z_1|x}^2}) N \left( \log \frac{1}{\sqrt{2\pi}s_1} - \frac{(x_1 - \gamma_1 z)^2}{2s_1^2} \right) dz \qquad (16)$$

$$= N \log \frac{1}{\sqrt{2\pi}s_1} - N \frac{\sqrt{s_{z_1}}}{2s_1^2} \left( \gamma_1^2 s_{z_1}^2 + x_1^2(1 - \gamma_1\gamma_{z_1}) \right) \qquad (17)$$

where the exponentiation by $N$ is due to the $N$ identical copies that are reconstructed using the same parameters. Due to the diagonal assumptions in the prior and variational approximation, the KL divergence breaks into two parts:

$$-KL[q_\phi(z|\{x_1\}, \{x_2\}, t, y)||p(z_1)]$$

$$= -\int \int \mathcal{N}(z_1|\gamma_{z_1}x_1, s_{z_1|x}^2)\mathcal{N}(z_2|\gamma_{z_2}x_2, s_{z_2|x}^2) \log \left( \frac{p(z_1)p(z_2)}{\mathcal{N}(z_1|\gamma_{z_1}x_1, s_{z_1|x}^2)\mathcal{N}(z_2|\gamma_{z_2}x_2, s_{z_2|x}^2)} \right) dz_1 dz_2$$

$$\qquad (18)$$

$$= -KL[\mathcal{N}(z_1|\gamma_{z_1}x_1, s_{z_1|x}^2)||p(z_1)] - KL[\mathcal{N}(z_2|\gamma_{z_2}x_2, s_{z_2|x}^2)||p(z_2)] \qquad (19)$$

where the term relevant for parameters regarding $z_1$ and $x_1$ is

$$-KL[\mathcal{N}(z_1|\gamma_{z_1}x_1, s_{z_1|x}^2)||p(z_1)] = -\log \frac{1}{s_{z_1|x}} - \frac{s_{z_1|x}^2 + \gamma_{z_1}x_1^2}{2} + \frac{1}{2} \qquad (20)$$

Bringing the two terms together and restricting the parameter space further so that $\gamma_1\gamma_{z_1} = 1$ and $s_1^{\frac{4}{5}} = s_{z_1|x} = s$, we have

$$\mathbb{E}_{q_\phi(z_1|x,t,y)}[\log p_\theta(x_1|z)] - KL[\mathcal{N}(z_1|\gamma_{z_1}x_1, s_{z_1|x}^2)||p(z_1)]$$

$$= N \log \frac{1}{\sqrt{2\pi}s_1} - N \frac{\sqrt{s_{z_1}}}{2s_1^2} \left( \gamma_1^2 s_{z_1}^2 + x_1^2(1 - \gamma_1\gamma_{z_1}) \right) - \log \frac{1}{s_{z_1|x}} - \frac{s_{z_1|x}^2 + \gamma_{z_1}x_1^2}{2} + \frac{1}{2}$$

$$\qquad (21)$$

$$= -\frac{5N}{8} \log s - N\frac{\gamma_1^2}{2} + \log s - \frac{s^2}{2} + \text{constants} \qquad (22)$$

Let's now consider the two scenarios where $N = 1$ and $N = 2$ we let $s \to 0$.

**N=1** The sum of the relevant, non-constant terms in the limit approaches minus infinity:

$$\lim_{s \to 0} \left( -\frac{5}{8} \log s - \frac{\gamma_1^2}{2} + \log s - \frac{s^2}{2} \right) = \lim_{s \to 0} (\frac{3}{8} \log s) + 0 = -\infty \qquad (23)$$

Thus, with no copies this approach clearly doesn't maximize the ELBO.

**N=2** The sum approaches infinity in the limit:

$$\lim_{s \to 0} \left( -\frac{10}{8} \log s - \gamma_1^2 + \log s - \frac{s^2}{2} \right) = \lim_{s \to 0} (-\frac{1}{4} \log s) + 0 = +\infty \qquad (24)$$

Although we only focused on a single observation in the ELBO, the final expression is not actually dependent on the values of $x_1$, so the ELBO will go to infinity for all observations with this parameterization. We can do the exact same thing for the second group of proxies $\{x_2\}$ and the second latent variable $z_2$. We conclude that while exactly this approach might not be the fastest way

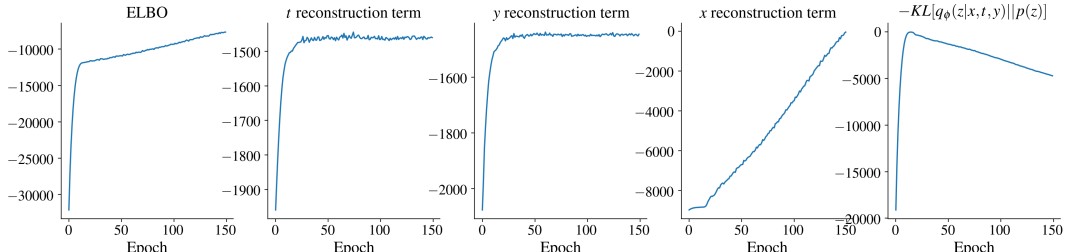

Figure 1: The ELBO and its parts for the linear-Gaussian data with exact proxies. The estimation model was the full, NN parameterized CEVAE. The data generating process was the same as in the main experiment, but without additional noise on the copies. The sample size was set to 1000, batch size was 200 and learning rate was 0.0001.

to increase the ELBO during training, it is clearly possible to get an infinite ELBO by using the latent space solely to reconstruct the proxies as accurately as possible. $\square$

In practice, the model could then follow a similar path as a result of gradient descent during training. Figure 1 shows the ELBO and its parts for a full, neural network parameterized CEVAE with copied proxies. While $t$ and $y$ reconstruction terms in the ELBO converge very early, the $x$ reconstruction term keeps improving long after that. The negative KL divergence term gets smaller, but it is not enough to counter the increase in $x$ reconstruction quality.

Intuitively, the reason that the final expression doesn't depend on the values of observed $x$ is that the latent space is able to represent the proxies perfectly, i.e., each $x_1$ is mapped to a corresponding latent representation through the $\gamma_{z_1}$ parameter. We restricted $\gamma_1\gamma_{z_1} = 1$ because if the latent representation of the proxy was scaled down or up by $\gamma_{z_1}$, we need to do the opposite scaling $\gamma_1 = \frac{1}{\gamma_{z_1}}$ in reconstruction.

Note that if the latent dimension is larger than two, we won't improve the ELBO by using them to reconstruct the treatment $t$ and effect $y$, as shown in Sec.D. Thus, the $p_\theta(t|z)$ and $p_\theta(y|z,t)$ reconstruction terms may only improve if the proxies, through the latent representation, are useful for predicting $y$ and $t$.

## D Proof of Proposition 3: Posterior collapse in the 1D linear CEVAE with no proxies

Here we show analytically that if the proxy reconstruction term is set to zero (essentially, we don't have any proxies), then a set of solutions where $p_\theta(y|do(t)) = p(y|t)$ are global maxima of the ELBO with linear-Gaussian data. These solutions correspond to situations where the latent space is not used in the reconstruction of $t$ or $y$, or either of them. In the proof, we assume that the estimation model is the CEVAE with a one-dimensional latent space and linearly parameterized conditionals. However, the result applies to a CEVAE with neural network parameterization as well if we assume that it can represent the same conditionals as the linear CEVAE.

**Proof.** Let's assume that we have maximized the ELBO so that $q_\phi(z|x,t,y) = p_\theta(z|t,y)$ for whatever $\theta$ that can maximize it. Then, with infinite data, according to Eq.10, we get that $p(t,y) = p_\theta(t,y)$. Let us use the notation $\gamma$ and $s$ to signify the CEVAE parameters that correspond to the parameters $c$ and $\sigma$ in the data generating model. In a similar way as in Sec.A we can then show that the inverse of the covariance matrix of $p_\theta(z,t,y)$ is then

$$
C^{-1} = \begin{bmatrix} 1 + \frac{\gamma_t^2}{s_t^2} + \frac{\gamma_{yz}^2}{s_y^2} & -\frac{\gamma_t}{s_t^2} + \frac{\gamma_{yz}\gamma_{yt}}{s_y^2} & -\frac{\gamma_{yz}}{s_y^2} \\ -\frac{\gamma_t}{s_t^2} + \frac{\gamma_{yz}\gamma_{yt}}{s_y^2} & \frac{1}{s_t^2} & -\frac{\gamma_{yt}}{s_y^2} \\ -\frac{\gamma_{yz}}{s_y^2} & -\frac{\gamma_{yt}}{s_y^2} & \frac{1}{s_y^2} \end{bmatrix}
$$

Inverting and marginalizing w.r.t. $z$, we then get

$$
C_{ty} = \begin{bmatrix} \gamma_t^2 + s_t^2 & \gamma_t^2\gamma_{yt} + \gamma_t\gamma_{yz} + \gamma_{yt}s_t^2 \\ \gamma_t^2\gamma_{yt} + \gamma_t\gamma_{yz} + \gamma_{yt}s_t^2 & \gamma_t^2\gamma_{yt}^2 + 2\gamma_t\gamma_{yt}\gamma_{yz} + \gamma_{yt}^2 s_t^2 + \gamma_{yz}^2 + s_y^2 \end{bmatrix}
$$

We have the equations

$$\mathrm{Var}(t) = \gamma_t^2 + s_t^2$$
$$\mathrm{Var}(y) = \gamma_t{}^2\gamma_{yt}{}^2 + 2\gamma_t\gamma_{yt}\gamma_{yz} + \gamma_{yt}{}^2 s_t{}^2 + \gamma_{yz}{}^2 + s_y{}^2$$
$$\mathrm{Cov}(t,y) = \gamma_t{}^2\gamma_{yt} + \gamma_t\gamma_{yz} + \gamma_{yt}s_t{}^2$$

This group of equations has many solutions, but two obvious groups of solutions stand out:

**Group 1.** $\gamma_{yz} = 0$, $s_y^2 = \mathrm{Var}(y) - \mathrm{Cov}(t,y)$ and $\gamma_{yt} = \frac{\mathrm{Cov}(t,y)}{\mathrm{Var}(t)}$.

This corresponds to the solution where $z$ doesn't have a direct causal effect on $y$, and thus there is no confounding and CEVAE doesn't use $z$ in reconstruction of $y$. We can also show that $p_\theta(y|do(t)) = p(y|t)$, by first calculating $p_\theta(y|do(t))$:

$$
\begin{aligned}
p_\theta(y|do(t)) &= \int p_\theta(y|z,t)p(z)dz \\
&= p_\theta(y|z,t) = p_\theta(y|t) \\
&= \mathcal{N}(y|\gamma_{yt}t, s_y^2) = \mathcal{N}(y|\frac{\mathrm{Cov}(t,y)}{\mathrm{Var}(t)}t, \mathrm{Var}(y) - \mathrm{Cov}(t,y)) \qquad (25)
\end{aligned}
$$

Here the second equality is true because $y$ is not dependent on $z$. The result corresponds to the conditional distribution formula for bivariate Gaussians: $p(y|t) = \mathcal{N}(y|\frac{\mathrm{Cov}(t,y)}{\mathrm{Var}(t)}t, \mathrm{Var}(y) - \mathrm{Cov}(t,y))$.

**Group 2.** $\gamma_t = 0$, $s_t^2 = \mathrm{Var}(t)$, $\gamma_{yz}^2 + s_y^2 = \mathrm{Var}(y) - \mathrm{Cov}(t,y)$ and $\gamma_{yt} = \frac{\mathrm{Cov}(t,y)}{\mathrm{Var}(t)}$.

This corresponds to the solution where $z$ doesn't have a direct causal effect on $t$, and thus again there is no confounding. Again, we can calculate $p_\theta(y|do(t))$:

$$
\begin{aligned}
p_\theta(y|do(t)) &= \int p_\theta(y|z,t)p(z)dz = \int \mathcal{N}(y|\gamma_{yz}z + \gamma_{yt}t, s_y^2)\mathcal{N}(z|0,1)dz \\
&= \mathcal{N}(y|\gamma_{yt}t, \gamma_{yz}^2 + s_y^2) = \mathcal{N}(y|\frac{\mathrm{Cov}(t,y)}{\mathrm{Var}(t)}t, \mathrm{Var}(y) - \mathrm{Cov}(t,y)) \qquad (26)
\end{aligned}
$$

The third equality was obtained with standard integration. This also corresponds to the conditional distribution formula for bivariate Gaussians, $p_\theta(y|do(t)) = p(y|t)$. $\square$

Other solutions to the group of equations are possible in principle, but in practice, the training usually converges to a solution similar to these ones, as witnessed in the repeated proxy experiment when loss scaling was set to zero. To take a closer look at these solutions, Fig. 2 visualizes the dependence of $y$, $t$, and $z$ for the trained models. For all of the models, $y$ is somewhat dependent on the $z$ (nonzero values of $\gamma_{yz}$), although not as much as it is on $t$. The treatment $t$, on the contrary, is almost not dependent at all on $z$, implying that the models correspond to solution group 2. In models 2 and 7, however, $z$ does affect the treatment $t$ a small amount as well, and these probably don't match that well with the analytical solutions explained above.

Note that while the proof applies strictly speaking only to the 1D linear CEVAE, the result is true for a neural-network parameterized CEVAE as well if it is able to represent the same conditional distributions. The solution is a global optimum with both parameterizations since already $p_\theta(t,y) = p(t,y)$, and thus it's not possible to improve the ELBO according to Eq.10.

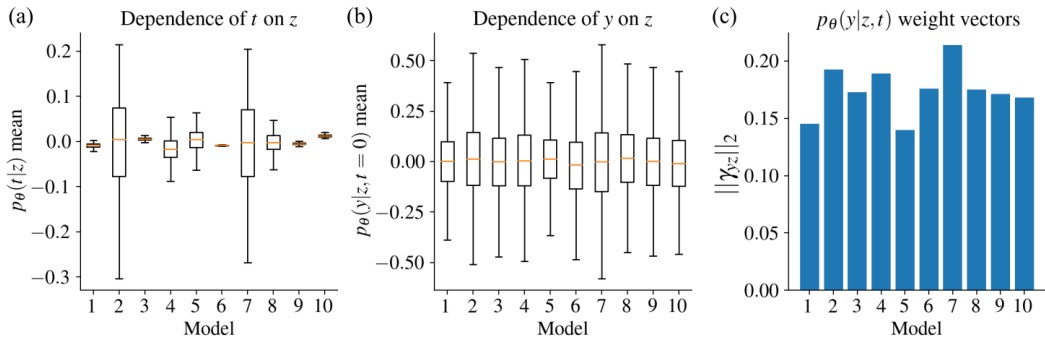

Figure 2: Visualizations of the latent space use of the ten models trained with proxy loss scaling equal to zero for the data with repeated proxies. (a) Means of $p_\theta(t|z)$ distributions with 10000 samples of $z$ from the prior of CEVAE. Aside from models 2 and 7, the mean is almost not dependent on $z$ at all. (b) Corresponding means of $p_\theta(y|z,t)$ distributions with 10000 samples of $z$ with $t = 0$. Clearly, $y$ is somewhat dependent on $z$ in all of the models. (c) The $L_2$ norms of the $z$ weight vectors of the $p_\theta(y|z,t)$ predictors. (The $p_\theta(y|z,t)$ predictors were linear and the latent spaces were 10-dimensional in this experiment.) Although $y$ is dependent on $z$ in all of the models, the $z$ weight vectors $\gamma_{yz}$ are much smaller than $\gamma_{yt}$, which was around 1.24 for all models.

# E   Experiment details

## E.1   Computing equipment and time taken to run experiments

The experiments were performed with two computers: A desktop computer containing an Intel i5-6500 processor and an Nvidia GTX 970 graphics card, and a laptop containing an AMD A12-9720P processor. Most of the experiments E.2-E.8 in this section took at most a single night to run with the computing equipment, although the binary data experiment took approximately an entire day. The basic Linear-Gaussian and binary experiments were conducted with the laptop, while the others were done with the desktop computer. The graphics card was used for the proxy MNIST data set, while the processor of the desktop was used for the rest.

The code for running the experiments is provided in the following github page: `https://github.com/severi-rissanen/critical_look_causal_dlvms`.

## E.2   Linear-Gaussian data

**Data generating parameters** The generating parameters were sampled with the following process: First, all of the standard deviations $\sigma$ were generated from a Gamma(1,5) distribution. Then, the structural coefficients $c$ were got by first sampling from a Gamma(0.3,4) distribution, multiplying with the corresponding $\sigma$ and adding the result to $\sigma/2$, and uniformly randomly flipping to a negative value. This resulted in the ratio $\frac{c}{\sigma}$ being not too close to zero while keeping the absolute values of $\sigma$ and $c$ roughly in the scale of 1. Too low a ratio for proxy, for instance, would mean that the proxy would be effectively very uninformative, and could cause even the analytical methods to fail. The generated parameters for the main experiment were $c_1 = 1.03$, $c_2 = 1.47$, $c_{yz} = 0.71$, $c_{yt} = -0.62$, $\sigma_{x_1} = 0.65$, $\sigma_{x_2} = 0.96$, $\sigma_t = 1.25$ and $\sigma_y = 0.48$.

**Estimation models** The default setup for the full, neural network parameterized CEVAE was so that each conditional distribution was represented with a three-layer MLP with a layer width of 30, using ELU activations. The (standard) assumption in the parameterization was that the outputs are normally distributed with a diagonal covariance for each network in the encoder and decoder. Thus, the final layers had twice the amount of heads than the output dimension, one for each mean and one for each standard deviation. There were four networks: The encoder, the proxy generation network ($p_\theta(x|z)$), the $t$ generation network ($p_\theta(t|z)$) and the $y$ generation network ($p_\theta(y|z,t)$). The dimension of the latent space was 10 for the default model. In the linear versions of CEVAE, the conditional distributions were defined with simple linear layers, and the standard deviations were separate, shared parameters used for all inputs.

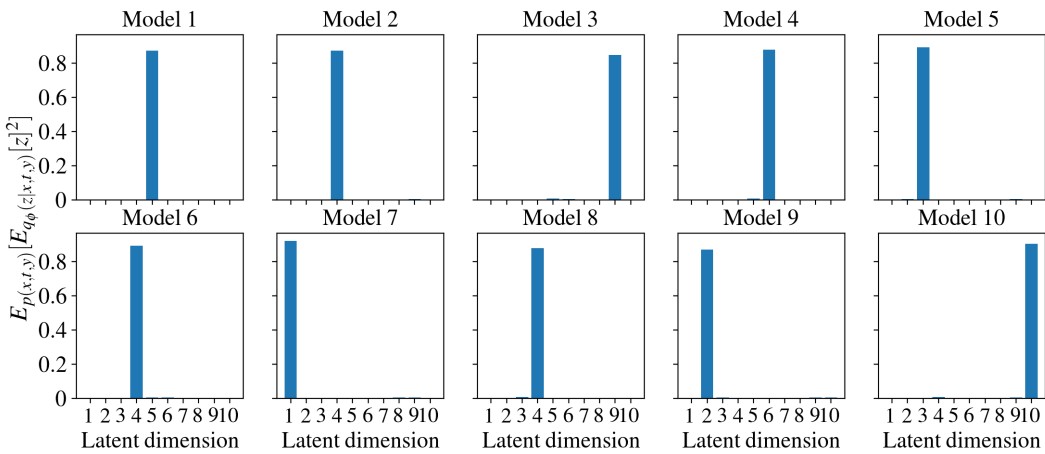

Figure 3: The squared expected values of the posterior approximations for the full 10D CEVAE with linear-Gaussian data, averaged over their respective data sets. Each of the models was trained on one of the linear-Gaussian data sets of size 20000.

**Training** The Adam optimizer was used for all models in this work. The neural network-based models were trained with 300 epochs, as that provided good loss function convergence for all data sizes. For the linear models, we used 500 epochs. The learning rates were annealed exponentially from 0.01 to 0.001. The batch size was 200. 10 data sets were sampled for each data size, and the models were trained once for each data set. The results from training to them provide the box plots in the results.

**Posterior collapse** Figure 3 visualizes the posterior collapse phenomenon for the 10 models trained with a data size of 20000. It shows the squared expected values $\mathbb{E}[q_\phi(z|x,t,y)]^2$ for each of the 10 dimensions, averaged over the data set (in other words, the variances of the encoder means for the data set). We see that all of the models use only one of the latent dimensions, while all the unused dimensions in the posterior approximation fall back to the mean of the prior.

**Failed 2D estimation** In the linear, 2D CEVAE experiment with failed estimation, we tried to initialize the model so that the parameters included aspects of the "correct" parameters, but were also sufficiently different to lead the model to incorrect estimation after training. The chosen initialization was also not itself a minimum of the loss function. Figure 5a shows the initialization. We tried to be very careful with the training by increasing the batch size to 1000, setting the learning rate to 0.001 and training until the model appeared to converge. In Fig.5b-c we plot the losses and estimates for the $c_{yt}$ coefficients as the training progressed. While the custom initialization results in an indistinguishable loss, the resulting causal effect is clearly wrong.

**Experiment with 10D linear CEVAE with an attempt to avoid posterior collapse** To take a deeper dive into the effect of posterior collapse on causal effect estimation with CEVAE, we designed an experiment with the purpose of maximizing disentanglement in the latent space of CEVAE. Before, a 2D linear CEVAE was somewhat prone to not posterior collapse completely, so we tried increasing the latent dimensionality to ten. We also annealed the KL divergence term from a low value to the regular one during training to promote disentanglement.

Figure 4a shows the results for a sample of of size 2000. The subpanel shows the scaling of the KL divergence term. We trained a 10D linear CEVAE twenty times, and a 1D linear CEVAE for comparison. The ten-dimensional model doesn't estimate the causal effect correctly, and instead, the estimates converge towards random values. The one-dimensional model, on the other hand, works correctly and settles on the correct value soon after the KL divergence term is returned to normal. From panel b we see that the ten-dimensional models indeed do use more than one dimension in reconstruction, similar to the situation with the failed 2D linear CEVAE in the main text. Importantly, the loss function at the end of training for the correct, one-dimensional model is indistinguishable from the ones of failed, higher-dimensional models. Thus, it's plausible that all of the models ended up in a global minimum or in a state very close to being one, and the model is unidentifiable with respect to causal effect estimates.

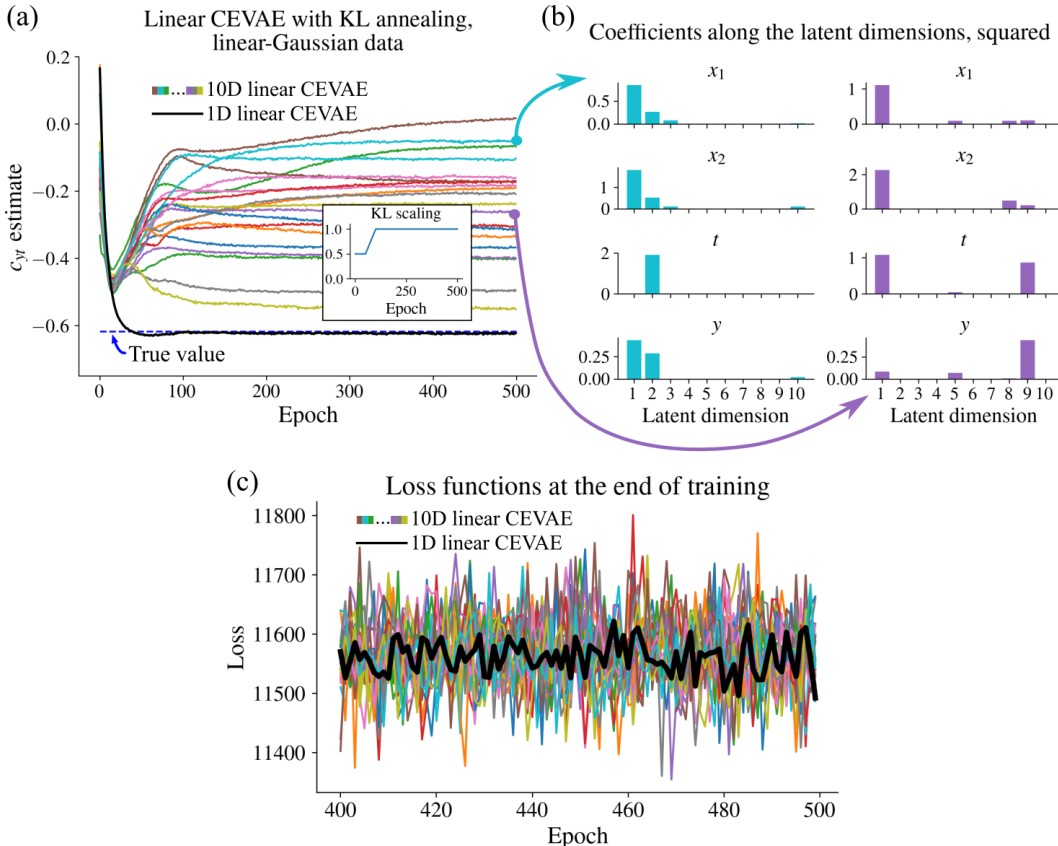

Figure 4: Results of the experiment where we try to discourage posterior collapse by annealing the KL divergence term from a low value at the beginning of training. All runs use a linear-Gaussian data set of size 2000. (a) $c_{yt}$ estimates for a linear CEVAE with a ten-dimensional latent space, trained twenty times, and for a linear CEVAE with a one-dimensional latent space. The 10D models do not estimate the causal effect correctly. (b) Coefficients of different linear predictors in the decoders of two selected models. Both use more than one dimension. (c) Loss functions towards the end of training. The 1D linear CEVAE doesn't reach a lower loss than the models that failed at estimation.

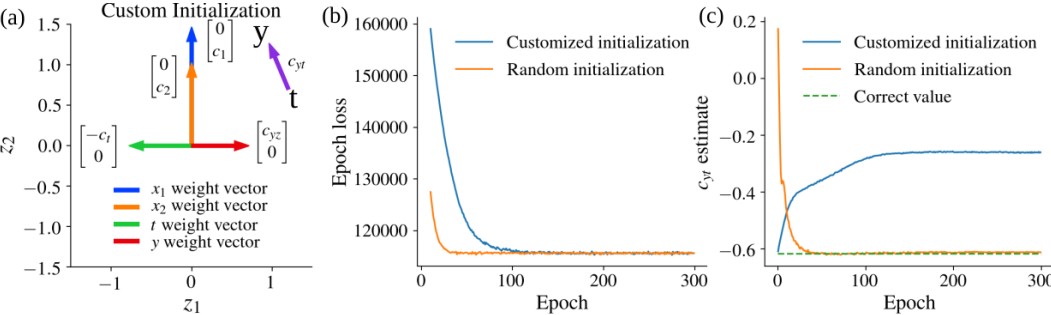

Figure 5: (a) Custom initialization of the 2D linear CEVAE that results in the wrong causal effect. The variances were set to the correct values and the posterior approximation was set so that it was the true posterior given the initialized decoder parameters. (b) Epoch losses of the custom initialized CEVAE and a successful, randomly initialized 2D linear CEVAE for comparison. (b) Estimates of $c_{yt}$, taken as the coefficient of $t$ in the linear predictor for $y|z, t$ in the decoder.

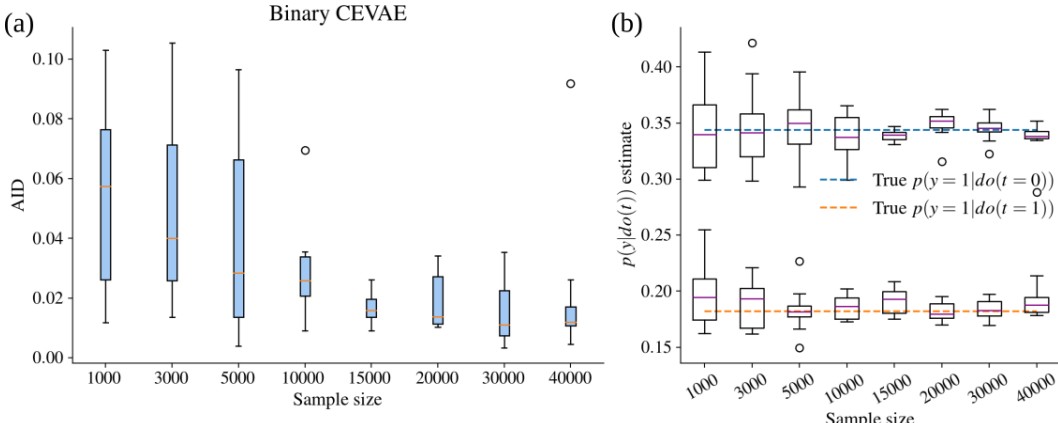

Figure 6: The AID and $p(y|do(t))$ estimates as a function of sample size for the binary CEVAE and binary data.

### E.3  Binary data

**Data generating parameters** The data generating parameters were generated from a Dirichlet(2) distribution for the prior $p(z)$ as well as all the conditional distributions. The distribution was chosen so that probabilities close to 0 or 1 would be unlikely, as too high or low probabilities can result in a pathological case where the data is very difficult to estimate from. The parameters for the main experiment were $p(z = 1) = 0.56$, $p(x_1 = 1|z = 0) = 0.56$, $p(x_1 = 1||z = 1) = 0.73$, $p(x_2 = 1|z = 0) = 0.94$, $p(x_2 = 1|z = 1) = 0.26$, $p(t = 1|z = 0) = 0.71$, $p(t = 1|z = 1) = 0.16$, $p(y = 1|z = 0, t = 0) = 0.57$, $p(y = 1|z = 0, t = 1) = 0.36$, $p(y = 1|z = 1, t = 0) = 0.17$ and $p(y = 1|z = 1, t = 1) = 0.04$.

**Estimation models** The default setup for the full CEVAE was the same as in the Linear-Gaussian experiment, except that the neural network heads for the decoder were transformed trough logistic link functions to probabilities, and the output was interpreted as a standard Bernoulli distribution. We also tried a model with a binary latent space, which was otherwise the same as the regular model, except the encoder had just one head, and the output was similarly interpreted as a Bernoulli distributed variable. The probability parameter of the Bernoulli distributed prior $p(z)$ was included as a learnable parameter. The expectation $\mathbb{E}_{q_\phi(z_i|x_i,t_i,y_i)}[\log p_\theta(x_i, t_i, y_i|z)]$ was calculated directly by passing both values of $z$ through the decoder weighting the log probabilities with $q_\phi(z_i|x_i, t_i, y_i)$.

**Training** The models were trained for 300 epochs with the Adam optimizer, with learning rate annealing from 0.01 to 0.0005, as that seemed to produce good loss function convergence. 10 data sets were sampled for each data size, and the batch size was 200.

**Binary CEVAE results**

Figure 6 shows the results from the binary CEVAE. It performs very well, and the causal effect estimates become better as the sample size is increased. However, the model doesn't work as well for all data generating processes, such as some of the ones in Sec. F.2, where it seemed that the model can get stuck in local minima easily.

### E.4  Linear-Gaussian data with redundant noise

**Data generation** The parameters for the first phase of the data generating distribution were $c_1 = 1$, $c_2 = 2$, $c_t = 0.5$, $c_{yz} = 0.6$, $c_{yt} = 1$, $\sigma_1 = 0.5, \sigma_2 = 0.7$, $\sigma_3 = 20$, $\sigma_t = 1$ and $\sigma_y = 1$ ($c_3$ was 0). After generating data from the linear-Gaussian distribution defined with these parameters, an additional orthogonal transformation was applied on the proxies so that the matrix was

$$R = \begin{bmatrix} \cos\alpha & -\sin\alpha & 0 \\ \sin\alpha & \cos\alpha & 0 \\ 0 & 0 & 1 \end{bmatrix} \begin{bmatrix} \cos\beta & 0 & \sin\beta \\ 0 & 1 & 0 \\ -\sin\beta & 0 & \cos\beta \end{bmatrix} \begin{bmatrix} 1 & 0 & 0 \\ 0 & \cos\gamma & -\sin\gamma \\ 0 & \sin\gamma & \cos\gamma \end{bmatrix} \quad (27)$$

where $\alpha = \frac{\pi}{4}$, $\beta = \frac{\pi}{4}$ and $\gamma = \frac{\pi}{4}$. Here $\alpha$, $\beta$ and $\gamma$ are also called the yaw, pitch and roll angles in the 3D space.

**Estimation models and training** The parameters of CEVAE and training were otherwise identical to the full CEVAE linear-Gaussian experiment, but the latent space was changed to one-dimensional and two-dimensional for the two setups explained in the main text.

### E.5 Linear-Gaussian data with copied proxies

**Data generation** The data generating parameters were $c_1 = 1$, $c_2 = 1$, $c_t = 0.5$, $c_{yz} = 0.6$, $c_{yt} = 1$, $\sigma_1 = 2$, $\sigma_2 = 2$, $\sigma_t = 1$ and $\sigma_y = 1$. $\tilde{x}_1$ and $\tilde{x}_2$ were copies of $x_1$ and $x_2$, with Gaussian noise of standard deviation 0.1 added.

**Estimation model** Other experimental parameters were exactly the same as for the full CEVAE in the linear-Gaussian experiment, except that we used a linear predictor for $y$ to make the results easier to interpret.

**Training** Training took only 100 epochs when trying out different loss scaling values, since that was clearly enough for convergence. The sample size was 20000 for the loss scaling experiment.

### E.6 Proxy IHDP data

**Consent and personally identifiable info.** The IHDP data is for public use and the data is not personally identifiable, as far as we are aware.

**Data generation** The data generating VAE was defined so that the encoder and decoder were MLPs with 3 layers and layer width 30 and a latent dimension of 10. The decoder was structured with the assumption that continuous variables had a Gaussian distribution at the outputs of the neural networks and categorical variables were categorically distributed with a softmax layer or logistic link function at the end for binary variables. The fourth variable in the IHDP data set was transformed to categorical since only four distinct values occured in the data set. Variances were estimated individually for each sample for the continuous variables with the neural networks. We chose the fifth variable as the unobserved confounder, and left the rest as the proxies. The treatment values were generated by first fitting a feed-forward neural network from the hidden confounder to the treatment values in the original data set and then sampling the treatments based on the probability given by the neural network. The $y$ values were then generated by $y^i = z^i + t^i \cdot \text{ATE} + \epsilon^i$, where $\epsilon^i$ is Gaussian noise with a standard deviation of 1 and ATE is the average treatment effect, which was also set to 1. ATE was set to 1 and the added Gaussian noise on $y$ had a standard deviation of 1. Figure 7 presents some aspects of the generating process: The chosen unobserved confounder is approximately normally distributed and it is clearly correlated with many of the observed variables. Panel (c) also visualizes the dependence of $t$ on $z$.

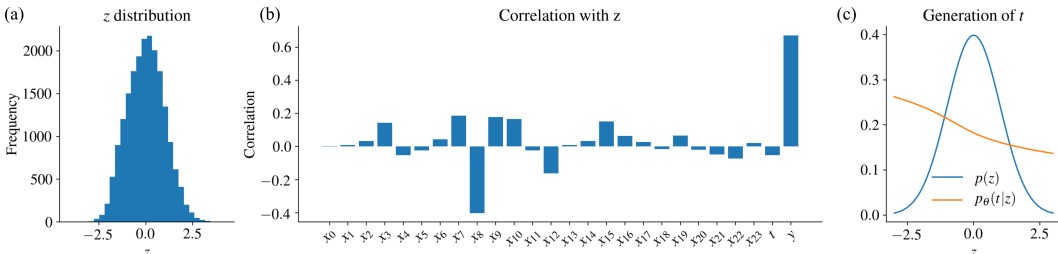

Figure 7: Properties of the IHDP data generating process. (a) The distribution of the chosen unobserved confounder. (b) Correlation of the chosen $z$ with the proxies, $t$ and $y$. (c) Visualization of the $p(t|z)$ function in the data generating process.

**Estimation model** The CEVAE model used for the data was defined similarly as the data generating VAE, with the addition of $t$ and $y$ prediction networks in the decoder. Another difference was that there were two separate networks in the decoder for $y$ generation that were chosen based on the value of $t$ on each pass. Likewise, the encoder consisted of four parallel networks that were chosen based on the value combination of $t$ and $y$ for each sample. The aim was to force the model to take $t$ and $y$

in to account in the reconstruction process, in a similar way as in the original CEVAE publication. Also, the fifth variable in the IHDP data naturally wasn't included in the proxies.

**Training** Training was done for 200 epochs with a batch size of 200 and exponential learning rate annealing from 0.001 to 0.00001. The sample size for the loss scaling experiment was 20000.

**Additional results** Figure 8 shows the experimental results for $\mathbb{E}[y|do(t = 0)]$. We see that the conclusions are the same as for $\mathbb{E}[y|do(t = 1)]$, which was presented in the main text.

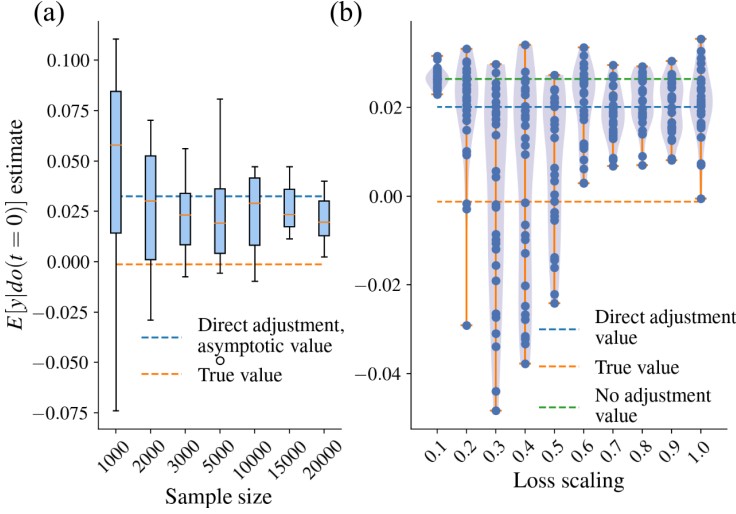

Figure 8: The $\mathbb{E}[y|do(t = 0)]$ estimates with respect to sample size and loss scaling with sample size set to 20000.

### E.7    Proxy MNIST data

**Consent and personally identifiable info.** The MNIST data is open access and the data is not personally identifiable, as far as we are aware.

**Data generation** The data generating GAN had a simple four-layer convolutional architecture in the discriminator and similarly four transpose convolutional layers in the generator. The exact details are in the code accompanying the article. The $t$ and $y$ values were Bernoulli distributed with a probability dictated by the value of the unobserved confounder (the first latent dimension of the GAN) through a logistic link function:

$$t|z \sim \mathrm{Bern}(\sigma(a_t z + b_t))$$
$$y|z, t \sim \mathrm{Bern}(\sigma(a_{y,1} z + b_{y,1})t + \sigma(a_{y,0} z + b_{y,0})(1 - t))$$

where $a_t = 1$, $b_t = 0.5$, $a_{y,1} = 2$, $b_{y,1} = 0.5$, $a_{y,0} = 2$ and $b_{y,0} = -0.5$. The additional linear-Gaussian proxy was generated from the unobserved confounder using $c = 1$ and $\sigma = 1$.

**Estimation model** The CEVAE model had was structured in a similar way as the GAN, with four transpose convolutional layers in the image part of the decoder. The other neural networks for the extra linear-Gaussian proxy, $t$ and $y$ were three-layer MLPs of width 30. Two NNs were defined for $y$, chosen depending on the value of $t$, the attempt being to define the conditional distributions as well as possible for the task. The encoder consisted of three parts. First, we had the four transpose convolutional layers starting from the images and ending up with 40 outputs. Second, we had a set of four fully connected linear layers with the 40 previous outputs and the additional proxy variable as inputs, with an output size of 125. One of the four networks was then chosen on each pass based on the value combination of $t$ and $y$. The third part was similarly a set of 4 fully connected linear layers with input size 25 and output size 40, where one of the networks was again chosen based on the values of $t$ and $y$. Again, the aim was to force the encoder to take $t$ and $y$ in to account. The final output was used to define the means and variances of the variational approximation in the 20-dimensional latent space. The convolutional and transpose convolutional layers were different from the corresponding GAN layers in that some of the kernel sizes, strides and paddings were changed around just so that the estimation model didn't match exactly with the data generation model.

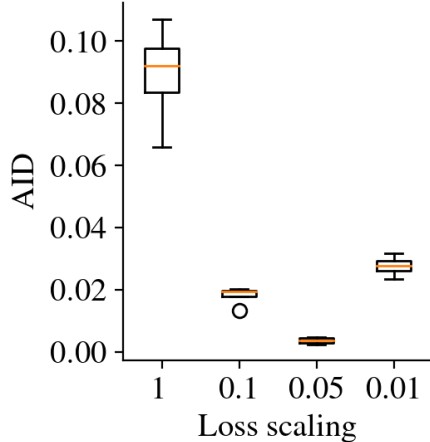

Figure 9: The AID values got by running CEVAE multiple times for each image reconstruction loss scaling factor. The data set was the same data set of size 10000 that was used to produce the figure in the main text.

**Training** The models were trained for 500 epochs (as shown in the Figures in the main text) with a batch size of 1000 and exponential schedule learning rate annealing from 0.003 to 0.001.

**Repeated experiment** Figure 9 shows the AID values of the same experiment run multiple times for each image reconstruction loss scaling value. The results confirm the pattern noted in the main text: The causal estimates are far off with scaling equal to 1, but get closer as we reduce it to 0.1 and the AID approaches almost zero with scaling 0.05. Decreasing further, the AID starts increasing again, indicating that there is an optimal value between zero and one.

### E.8 Twins data

The original CEVAE paper presented the Twins data as an example of a data set where the causal effects are identifiable in principle (with knowledge of the data generating process), but the distribution of $z, t$, and $y$ is derived from real data. They reported that the model produced better estimates than a direct adjustment to the proxies, but our focus is on the asymptotic performance and the identifiability of the model. It is interesting from our point of view also because the unobserved confounder is categorical, and it provides us with a further example of a misspecified latent variable in addition to the binary data experiment.

**The data.** The data consists of information about twin births and infant mortalities in the USA during the years 1989-1991. The "treatment" is defined to be the variable that tells us whether the particular newborn was the heavier or lighter of the twins ($t = 0$ for the lighter twin, $t = 1$ for the heavier). The outcome $y$ is a binary variable that indicates mortality within a year after birth. There are 46 covariates relating to the parents, out of which the GESTAT10 feature is chosen as the unobserved confounder. It consists of ten categories indicating the number of gestation weeks before birth. An observational study is then simulated by choosing one of the $t$ values for each twin pair based on the value of the confounder: $\text{Bern}(\sigma(w_o^T \vec{v} + w_h(z/10 - 0.1)))$, where $\vec{v}$ is a vector that contains the other covariates. Having the other covariates in the formula produces some additional randomness to the assignment of $t$. The weight vectors $w_o$ and $w_h$ are sampled from $w_o \sim \mathcal{N}(0, 0.1 \cdot I), w_h \sim \mathcal{N}(5, 0.1)$. Proxies are created based on the categorical confounder by taking a one-hot representation, copying it three times, and randomly flipping each binary feature with some probability. The original paper used probabilities ranging from 0.05 to 0.5, and we use 0.2 in our experiments. We generated $w_o$ and $w_h$ once and used those in the experiments to reduce the amount of randomness. The used $w_h$ weight was 5.030. Table 1 lists the coefficients of the weight vector $w_h$.

Because we want to consider data sets that are larger than the original data set, we decided to use bootstrapping to generate new sets that are distributed according to the empirical distribution of the Twins data. So the full data generating process was that $z, v$, and both $y$ values were sampled with replacement for a given number of times from the initial data, after which the $t$ values and the proxies were sampled with the procedure explained earlier. The mortality of the

| $w_{o,0}$ | $w_{o,1}$ | $w_{o,2}$ | $w_{o,3}$ | $w_{o,4}$ | $w_{o,5}$ | $w_{o,6}$ | $w_{o,7}$ | $w_{o,8}$ | $w_{o,9}$ |
|---|---|---|---|---|---|---|---|---|---|
| 0.068 | 0.167 | -0.187 | 0.150 | -0.177 | -0.101 | 0.143 | 0.047 | 0.112 | -0.039 |
| $w_{o,10}$ | $w_{o,11}$ | $w_{o,12}$ | $w_{o,13}$ | $w_{o,14}$ | $w_{o,15}$ | $w_{o,16}$ | $w_{o,17}$ | $w_{o,18}$ | $w_{o,19}$ |
| -0.087 | -0.098 | -0.077 | -0.070 | -0.069 | -0.063 | -0.156 | -0.070 | -0.037 | 0.143 |
| $w_{o,20}$ | $w_{o,21}$ | $w_{o,22}$ | $w_{o,23}$ | $w_{o,24}$ | $w_{o,25}$ | $w_{o,26}$ | $w_{o,27}$ | $w_{o,28}$ | $w_{o,29}$ |
| 0.141 | -0.035 | 0.204 | 0.153 | -0.070 | 0.273 | -0.008 | -0.143 | 0.109 | -0.155 |
| $w_{o,30}$ | $w_{o,31}$ | $w_{o,32}$ | $w_{o,33}$ | $w_{o,34}$ | $w_{o,35}$ | $w_{o,36}$ | $w_{o,37}$ | $w_{o,38}$ | $w_{o,39}$ |
| 0.127 | -0.017 | 0.005 | 0.077 | 0.081 | 0.019 | -0.026 | -0.078 | -0.172 | -0.051 |
| $w_{o,40}$ | $w_{o,41}$ | $w_{o,42}$ | $w_{o,43}$ | $w_{o,44}$ | $w_{o,45}$ | | | | |
| 0.114 | -0.134 | 0.018 | -0.104 | -0.132 | 0.123 | | | | |

Table 1: The $w_o$ weight vector used in the Twins data experiments. It was sampled from the distribution $\mathcal{N}(0, 0.1 \cdot I)$.

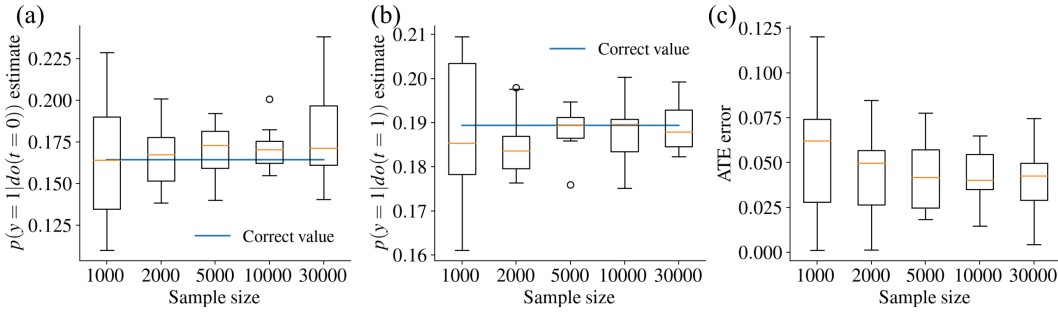

Figure 10: Causal effect estimates and the ATE errors for the Twins data set.

lighter twin is $p(y = 1|do(t = 0)) = 18, 9\%$ whereas the heavy twin mortality is slightly lower, $p(y = 1|do(t = 1))) = 16, 4\%$.

**Consent and personally identifiable info.** The NCHS data the Twins data set is based on is for public use and the data is not personally identifiable, as far as we are aware.

**Estimation model.** The used CEVAE model was the regular full CEVAE, with the exception that layer width was 50 instead of 30 in the encoder. We made this choice because the number of observed variables was over 30, and using the regular width would have acted as a bottleneck and could have caused sub-optimal results.

**Training.** In the experiment with varying sample sizes and multiple different data sets, the models were trained for 1000 epochs with a batch size of 500 and learning rate annealing going from 0.004 to 0.0002. With the experiments where the model was trained multiple times on one data set, the number of epochs was 2000 to make sure that the estimates seemed to converge to some value.

**Results.** Figure 10 shows the causal effect estimates and the ATE error as a function of sample size. While the ATE errors are roughly in the same order of magnitude as the ones reported in the original paper for proxy noise level 0.2, it's not clear that the estimates converge towards the correct values. In fact, it looks like they are not converging towards any value in particular, even though quite a lot of effort was put into careful training. To take a closer look at this issue, we trained the model multiple times for single generated data sets of sizes 10000 and 50000 and recorded the causal effect estimates as the training progressed. Figure 11 shows the results for both data sets. With a sample size of 10000, the model isn't able to get a consistent estimate, and increasing the amount of data doesn't seem to help either, since the same happens with a sample size of 50000. It seems that the model ends up in different kinds of minima that produce different kinds of estimates. Figure 12 shows that using the loss value at the end of training can not be used to choose a well-performing model, as models that got (slightly) smaller losses than others produce even worse estimates than many models with a higher loss. The losses are also very close to each other, so one could say that this is a case of "effective" model unidentifiability, where it's not possible to find a single estimate.

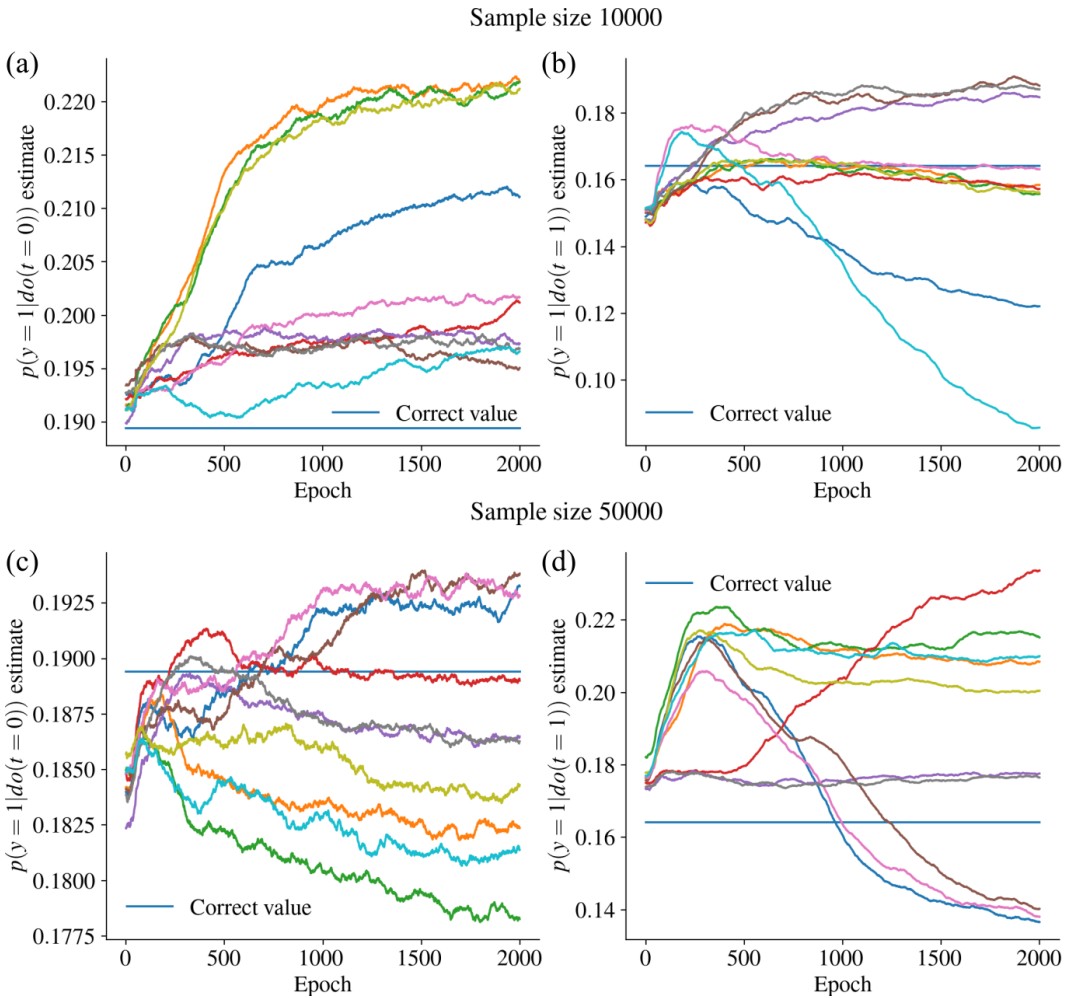

Figure 11: Causal effect estimates with respect to the number of epochs during training for the Twins data. The estimates are averaged over a window of 100 epochs to smooth out temporary variation. (a),(b) Ten runs for one data set of size 10000. (c),(d) Ten runs for one data set of size 50000.

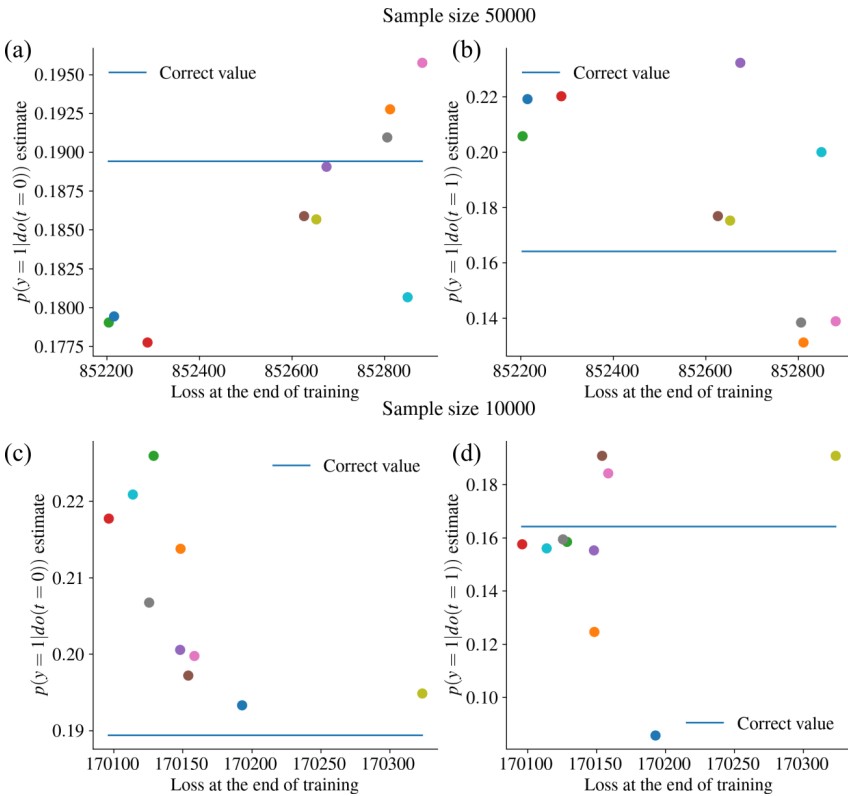

Figure 12: The causal effect estimates with respect to the loss at the end of training for the Twins data set. The losses and estimates are averages over the final one hundred epochs. (a),(b) Estimates for a data set of size 10000. (c),(d) Estimates for a data set of size 5000.

# F  Replicated experiments

## F.1  Linear-Gaussian data

We sampled four other data-generating distributions with the process detailed in Section E.2 and ran the same experiment comparing the convergence of AID values on each process. The parameters are presented in Table 3. Figure 13 shows the results. While the convergence of AID values to zero happens differently for different data generating processes, the main conclusion stays the same. The estimates of the full CEVAE model are usually slightly less accurate than the 1D linear CEVAE and the analytical method, but they do steadily improve as the sample size increases.

Table 2: Data generating parameters for the repeated linear-Gaussian experiments.

|  | $c_1$ | $c_2$ | $c_t$ | $c_{yz}$ | $c_{yt}$ | $\sigma_1$ | $\sigma_2$ | $\sigma_t$ | $\sigma_y$ |
|---|---|---|---|---|---|---|---|---|---|
| Process 1 | -0.53 | 0.92 | 0.99 | -1.15 | 0.46 | 0.71 | 1.02 | 1.14 | 0.84 |
| Process 2 | 1.05 | -0.57 | -0.83 | 0.76 | -1.38 | 1.04 | 0.77 | 0.68 | 1.11 |
| Process 3 | 1.30 | -1.02 | 0.80 | 1.17 | 1.11 | 1.02 | 0.91 | 1.27 | 0.88 |
| Process 4 | -1.58 | 0.80 | -0.82 | 0.99 | -1.13 | 1.28 | 0.87 | 1.04 | 0.77 |

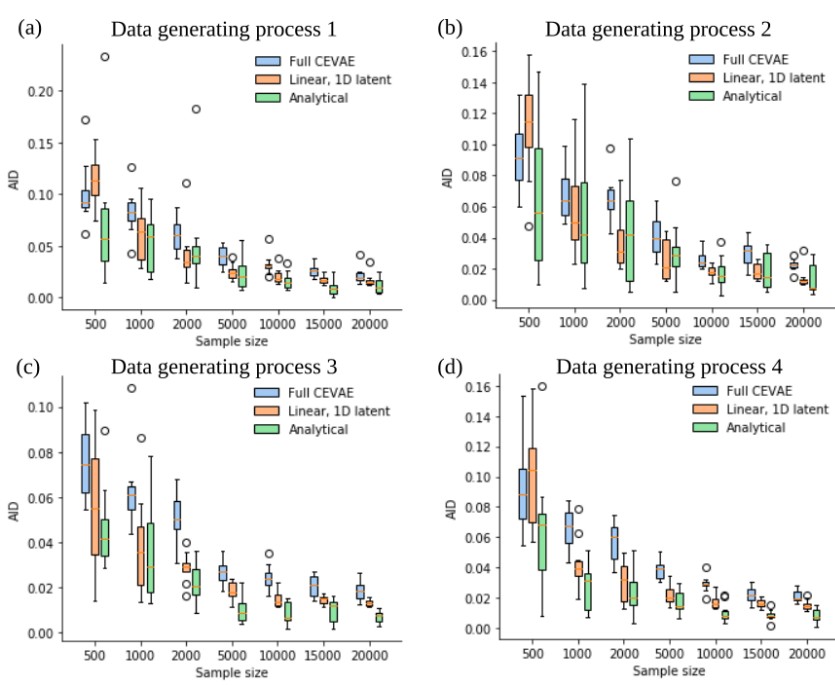

Figure 13: The AID values with respect to sample size for the replicated linear-Gaussian experiments.

## F.2  Binary data

Four other data generating processes were sampled from the distribution explained in Sec.E.3, and the same experiment as in the main text was run for each data generating process. The results are presented in Fig.14. Aside from possibly the second data set, CEVAE consistently estimates the causal effects incorrectly. The analytical method is not very accurate with small sample sizes, but the average estimate is correct, in contrast to CEVAE. The fact that CEVAE performed well especially on the second data generating process may be due to the true causal effects being very close to the effects we get without adjustment at all (compare $p(y = 1|do(t = 0)) = 0.468$ to $p(y = 1|t = 0) = 0.473$ and $p(y = 1|do(t = 1)) = 0.138$ to $p(y = 1|t = 1) = 0.136$).

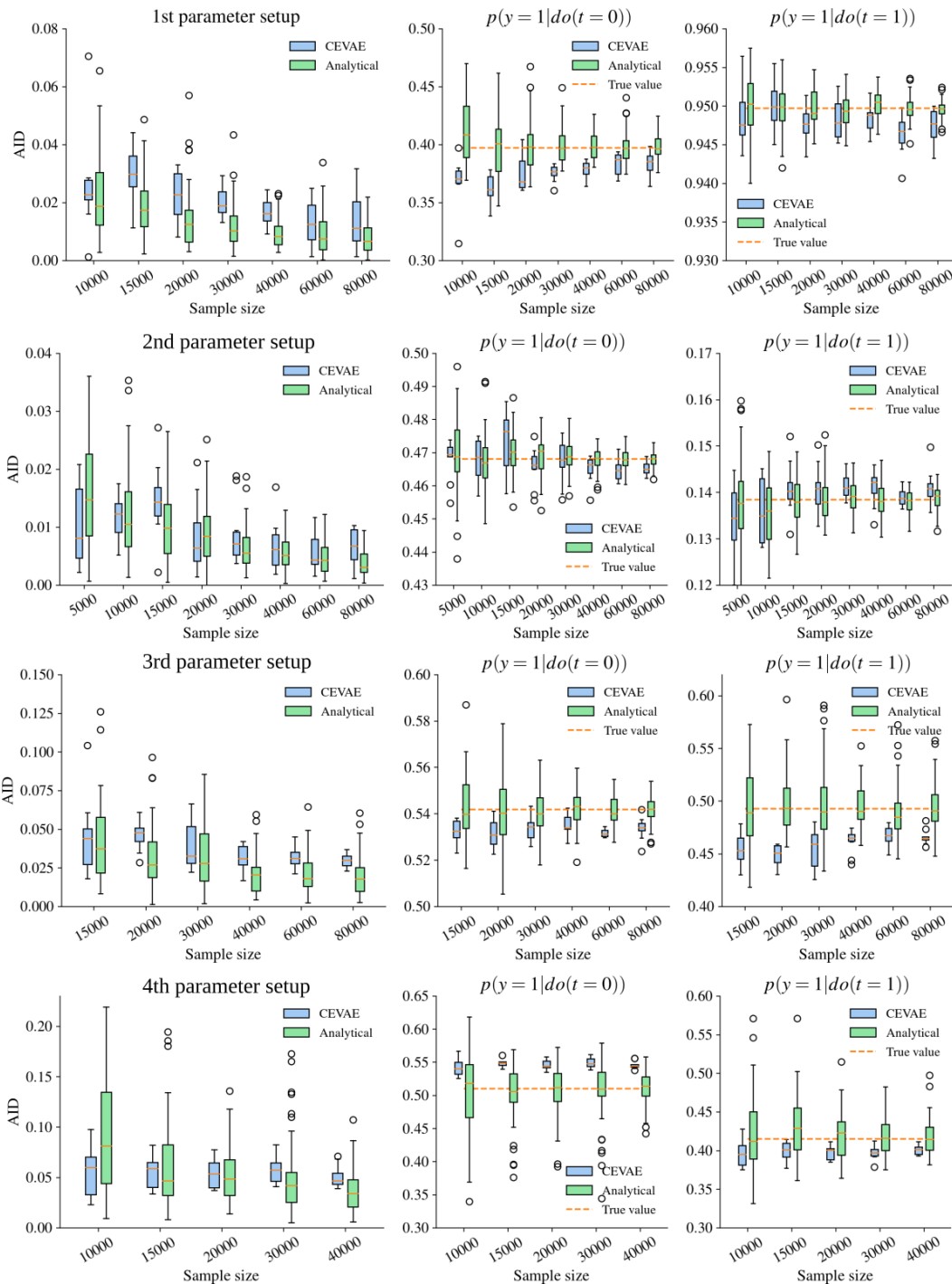

Figure 14: The results from the four additional binary data generating processes. Some outliers are not shown in order to make the plots readable.

Table 3: Data generating parameters for the repeated binary data experiments.

|  | Process 1 | Process 2 | Process 3 | Process 4 |
|---|---|---|---|---|
| $p(z = 1)$ | 0.41 | 0.49 | 0.42 | 0.45 |
| $p(x_1 = 1 \mid z = 0)$ | 0.88 | 0.24 | 0.63 | 0.30 |
| $p(x_1 = 1 \mid z = 1)$ | 0.66 | 0.73 | 0.44 | 0.44 |
| $p(x_2 = 1 \mid z = 0)$ | 0.63 | 0.53 | 0.81 | 0.33 |
| $p(x_2 = 1 \mid z = 1)$ | 0.86 | 0.63 | 0.47 | 0.65 |
| $p(t = 1 \mid z = 0)$ | 0.51 | 0.44 | 0.19 | 0.25 |
| $p(t = 1 \mid z = 1)$ | 0.78 | 0.29 | 0.64 | 0.78 |
| $p(y = 1 \mid z = 0, t = 0)$ | 0.21 | 0.42 | 0.49 | 0.67 |
| $p(y = 1 \mid z = 0, t = 1)$ | 0.93 | 0.12 | 0.72 | 0.54 |
| $p(y = 1 \mid z = 1, t = 0)$ | 0.66 | 0.52 | 0.61 | 0.31 |
| $p(y = 1 \mid z = 1, t = 1)$ | 0.97 | 0.15 | 0.18 | 0.26 |