# OpenReview forum: "A Critical Look at the Consistency of Causal Estimation with Deep Latent Variable Models"
_NeurIPS.cc/2021/Conference — NeurIPS 2021 Poster_

### Official Review · Reviewer_zjui · 2021-07-15

**Rating:** 6
**Confidence:** 3

**Summary:**

This paper investigated the consistency property of CEVAE, mainly under unknown parametric form of generative model (p(x|z),p(t|z),p(y|t,z)), the unknown distribution of latent variables, and correlated proxies. Theoretical analysis under some special cases is given. Most conclusions are based on empirical observations.

**Limitations And Societal Impact:**

The authors have adequately addressed the limitations and potential negative social impact of their work.

**Main Review:**

Although the observations of CEVAE are quite interesting, I am not sure whether this paper is beneficial to the broad audience since the analysis is only narrowed to one method (CEVAE) that is specific to the causal graph with unobserved confounder. Besides, some observations are not new, such as that the CEVAE can fail to recover the true causal effect with the incorrect specification of the distribution of the latent variables, which is not surprising given the non-identifiability of the VAE model for the latent variables. That is, whether ("\int p(z)p(y|t,z) dz" is determined by the p(y,t,x) that can be recovered by the generative model like VAE).

The observations with correlated proxies, and especially the scale factor analysis are interesting. I believe this result can benefit other works that implement VAE to learn representations. However, as I mentioned earlier, overall the perspective of this paper is somewhat narrow and not novel enough for publication.

Post-Rebuttal.

Change my score from 5 to 6. This is nice work.

**Time Spent Reviewing:**

3 hours

---

> ### Author Response · Authors · 2021-08-09
> **We argue that while our results may not seem surprising, they are useful to the field.**
>
> Thank you for the comments. Regarding the novelty of the paper:
>
> 1) To our knowledge, no study has been conducted on the identifiability or consistency of causal effect estimates using deep latent variable models. Note also that the identifiability of the causal effect estimate ("\int p(z)p(y|t,z) dz") is different from the identifiability of the latent representation.
>
> 2) We agree that in hindsight the results may not be surprising. Nonetheless, previously promising results have been published and similar approaches have gained popularity; hence, the results were not obvious to us. Additionally, the model performed surprisingly well in some cases, such as with the linear-Gaussian data.

---

### Official Review · Reviewer_6k6q · 2021-07-16

**Rating:** 8
**Confidence:** 3

**Summary:**

The paper presents a number of interesting analysis and empirical findings about the behavior of the CEVAE model when it is employed to estimate causal effects when there are proxies observations available for the unobserved confounders. The paper presents several insights about when and why the CEVAE model fails or succeeds, including settings close to real-world use cases.

**Limitations And Societal Impact:**

No problems.

**Main Review:**

The paper contributes valuable insights to the  understanding of the limitations of CEVAE methods, especially what can go wrong in real-world usage. I found one of the key insights being that the scaling of the losses of CEVAE determines whether the right solution is found or not. However, this insight was only discovered by post-mortem analysis.

Therefore, I would be interested in the following questions to the authors:
- Would it be possible to propose a guideline or an algorithm how the scaling of the CEVAE loss can be accomplished?
- Would use of cross-validation solve this? Or some kind of normalization of the losses in the ELBO.

Found the paper easy to read and having a clear structure.

**Time Spent Reviewing:**

2

---

> ### Author Response · Authors · 2021-08-09
> **Ideas on choosing the correct scaling factor for CEVAE loss**
>
> Thank you for the comments and questions.
>
> Using cross-validation does not seem straightforward because the true causal effect is not available in the test data (but only the factual outcome). This is something to study more in the future, but one direction that seems worth exploring is whether influence functions that have been used as a replacement for C-V could be useful here [1].
>
> Alternatively, if we can assume that we have more information about the relationships between the proxies, or could learn those from the data, then we might be able to ignore repeated proxies and only use the remaining for the training of CEVAE, in which case scaling might not be necessary.
>
> References:
> [1]  Alaa, Ahmed, and Mihaela Van Der Schaar. "Validating causal inference models via influence functions." International Conference on Machine Learning. PMLR, 2019.

---

### Official Review · Reviewer_1bWo · 2021-07-18

**Rating:** 6
**Confidence:** 3

**Summary:**

The paper takes an in-depth look at using latent variable models in causal effect estimation from proxies. The idea behind such methods, specifically CEVAE, is that the proxies provide sufficient information about the confounder (one extreme case is when the proxies determine the confounder) to ensure that the causal effect is identified. As the authors suggest and show, both theoretically and empirically, this idea bears fruit in terms of estimability of causal effects only under potentially strong assumptions. The authors outline multiple failure modes of the CEVAE which I believe extend to other existing deep learning based methods for proxy-based effect estimation. Finally, experiments show in some more semi-synthetic datasets, the estimated effects are not consistent.

**Limitations And Societal Impact:**

Interesting and important paper, requires some clarifications about identification vs. consistency.

**Main Review:**

The paper addresses an important problem that prevails in a few existing methods (including CEVAE) for causal effect estimation from proxies. In general this problem is not solvable without further assumptions even when the models being used for encoder and decoder can be arbitrarily flexible. This flexibility is one of the culprits because, while the observed data has information about the confounder , this is insufficient to guarantee that the that model p( Y | Z, T ) built on treatment T and the latent variable Z captures the causal effect of T. This is discussed in the paper around proposition 2 where the model converges to using the proxies to adjust instead of the confounder.

In other cases, because the loss term is reconstructions, the confounding (non-causal) relationship in the treatment is included in the returned estimate of the causal effect, as in p(Y|T) instead of p(Y | do(T)).

A few points of lack of clarity still remain:

1. Causal identification with do-calculus makes no parametric assumptions about data generating mechanisms in full generality; the do-calculus methodology was developed to separate causal identification from statistical parameter identification (for parametric models like linear coefficients)
2. Posterior collapse means that the confounder is determined by the proxies, outcome, and treatment. This is an interesting situation because it seems to suggest that one can estimate the confounder in such a setting. However it is unclear how one can guarantee that p(X | Z) does not collapse and positivity (overlap) is violated. When overlap is violated, causal effect estimation is not possible in general without further, potentially strong, assumptions.
3. Quite a few experiments mention lack of consistency but this suggests the problem comes from not having enough data but not due to a failure of causal identification. Could the authors clarify which of these is the reason?

I would suggest the authors clarify which of the many failure modes presented are about causal identification and which are about statistical identification. For example, in the gaussian case, the failure seems to be statistical but in general, the failure is causal. Further, the discussion about identification should be separate from the discussion about consistency.

**Time Spent Reviewing:**

4

---

> ### Author Response · Authors · 2021-08-09
> **Answers regarding do-calculus, posterior collapse and failure modes**
>
> Thank you for the questions and comments. Answers to questions in order:
>
> 1) It is a good point that do-calculus can produce useful results by only assuming the causal graph, and our wording describing it in Section 2.2 may have been somewhat misleading. It is, however, sometimes necessary to make assumptions about the parametric form of the causal data generating process to make some effects identifiable in the causal sense, especially in the presence of unobserved variables. This is what we intended to communicate with the sentence that mentioned do-calculus. An example is the paper by Kuroki and Pearl, who showed that it is possible to identify the causal effect by assuming that the data generating process is linear-Gaussian. Regarding separation of causal and statistical identification, the parametric form of the statistical estimator is indeed different in our experiments, and here CEVAE is used as a highly flexible estimator. We will clarify this in the final version.
>
> 2) What we mean by posterior collapse here is that the posterior of the latent variable is the same as the prior for some of the latent dimensions (see, e.g. [1]). The overlap assumption concerns the data and hence it is not directly affected by posterior collapse or CEVAE. However, if the assumption is violated then the estimation indeed likely fails (estimates are expected to have a very large variance).
>
> 3) What we meant by the lack of consistency was that when we increased the amount of data in our experiments, we saw the causal effect converging to a wrong value. Hence, adding more data should not help. As we discuss in Section 2.2., estimation may fail because: 1) the causal effects are not identifiable in principle from the data 2) CEVAE as a statistical estimator is not identifiable (model identifiability) 3) CEVAE is model-identifiable, but the causal effect estimate is still wrong, e.g., because the model is wrongly specified (as is the case with the binary data). In practice, as we have mentioned in the text, all of the experiments are known to be identifiable in principle except for the real-data experiments in Section 3.3. Hence, the cause of failure is the model identifiability or other issues that result in CEVAE not being a consistent estimator of causal effects. We will clarify this.
>
> References:
> [1] Dai, Bin, Ziyu Wang, and David Wipf. "The usual suspects? Reassessing blame for VAE posterior collapse." International Conference on Machine Learning. PMLR, 2020.

---

### Official Review · Reviewer_DRMb · 2021-07-19

**Rating:** 6
**Confidence:** 3

**Summary:**

The paper investigates this gap between theory and empirical results with analytical considerations and experiments under multiple synthetic and real-world data sets, using the causal effect variational autoencoder (CEVAE) as a case study. According to their study, CEVAE seems to work reliably under some simple scenarios, it does not estimate the causal effect correctly with a misspecified latent variable or a complex data distribution, as opposed to its original motivation.

**Limitations And Societal Impact:**

1.    Based on the writing and experimental presentation, the paper is clear and well written.

2.    The paper focuses on simple causal models. Authors could give more discussion on multiple latent variables and multiple confounder cases.

3. Additional theoretical studies could be presented to make the paper more informative so that readers could learn more from the paper.


**Main Review:**

1.    The reviewer is not very familiar with the topic. But based on the draft, the authors present a solid study on the experimental gap of causal discovery with CEVAE.

2.    The paper focuses on empirical issues with CEVAE. However, most of these issues are intuitive, and not difficult to validate with experiments.  There are no surprising results or very deep technical theories along with the experiments.


**Time Spent Reviewing:**

8

---

> ### Author Response · Authors · 2021-08-09
> **Comments regarding additional theoretical studies**
>
> Thank you for the comments. While we didn’t present many theoretical results we hope that our work inspires further work on theoretical guarantees to causal inference using deep latent variable models. An interesting continuation to this work could be to investigate whether the full neural-network parameterized CEVAE is consistent on the linear-Gaussian data in the same way the linear CEVAE is. This could possibly be approached by enforcing some assumptions about the conditional distributions.

---

### Decision · Program_Chairs · 2021-09-27

**Decision:**

Accept (Poster)

**Comment:**

The paper takes a deep dive into issues of consistency and so-called "model identifiability" of CEVAE, a method for causal inference with proxy variables. The authors outline multiple possible failure modes of CEVAE, which could also extend to other existing deep learning based methods for proxy-based effect estimation and possibly even more generally to other latent variable models. These findings are validated over a set of generally well-designed experiments. This is one of only a few papers which look into the assumptions at the basis of using flexible general-purpose latent variable models such as VAEs for learning tasks which go beyond prediction. Further, it is the first to specifically address the problem of these assumption when working at the intersection of VAEs and causal inference, thus highlighting several points which should be of interest to all future work in the sub-field.

The reviewers brought up several points relating to the clarity of presentation which I trust the authors will address in the final version; in addition, I would appreciate a discussion of methods other than CEVAE for which some of the findings might be relevant.